**Title: Simultaneous disintegration of outlet glaciers in Porpoise Bay (Wilkes Land), East Antarctica, driven by sea-ice break-up.**

**Authors:** B.W.J. Miles[1*], C. R. Stokes[1], S.S.R. Jamieson[1]

**Affiliation:** [1]*Department of Geography, Durham University, Science Site, South Road, Durham, DH1 3LE,*

*UK*

*\*Correspondence to: a.w.j.miles@durham.ac.uk*

**Abstract: The floating ice shelves and glacier tongues which fringe the Antarctic continent are important because they help buttress ice flow from the ice sheet interior. Dynamic feedbacks associated with glacier calving have the potential to reduce buttressing and subsequently increase ice flow into the ocean. However, there are few high temporal resolution studies on glacier calving, especially in East Antarctica. Here we use ENVISAT ASAR wide swath mode imagery to investigate monthly glacier terminus change across six marine-terminating outlet glaciers in Porpoise Bay (-76°S, 128°E), Wilkes Land (East Antarctica), between November 2002 and March 2012. This reveals a large near-simultaneous calving event in January 2007, resulting in a total of ~2,900 km$^2$ of ice being removed from glacier tongues. We also observe the start of a similar large near-simultaneous calving event in March 2016. Our observations suggest that both of these large calving events are driven by the break-up of the multi-year sea-ice which usually occupies Porpoise Bay. However, these break-up events appear to have been driven by contrasting mechanisms. We link the 2007 sea-ice break-up to atmospheric circulation anomalies in December 2005 weakening the multi-year sea-ice through a combination of surface melt and a change in wind direction prior to its eventual break-up in January 2007. In contrast, the 2016 break-up event is linked to the terminus of Holmes (West) Glacier pushing the multi-year sea-ice further into the open ocean, making the sea-ice more vulnerable to break-up. In the context of predicted future warming and the sensitivity of sea-ice to changes in climate, our results highlight the importance of interactions between landfast sea-ice and glacier tongue stability in East Antarctica.**

# 1. Introduction

Iceberg calving is an important process that accounts for around 50% of total mass loss to the ocean in Antarctica (Depoorter et al., 2013; Rignot et al., 2013). Moreover, dynamic feedbacks associated with retreat and/or thinning of buttressing ice shelves or floating glacier tongues can result in an increased discharge of ice into the ocean (Rott et al., 2002; Rignot et al., 2004; Wuite et al., 2015; Fürst et al., 2016). At present, calving dynamics are only partially understood (Benn et al., 2007; Chapuis and Tetzlaff, 2014) and models struggle to replicate observed calving rates (van der Veen, 2002; Astrom et al., 2014). Therefore, improving our understanding of the mechanisms driving glacier calving and how glacier calving cycles have responded to recent changes in the ocean-climate system is important in the context of future ice sheet mass balance and sea level.

Calving is a two-stage process that requires both the initial ice fracture and the subsequent transport of the detached iceberg away from the calving front (Bassis and Jacobs, 2013). In Antarctica, major calving events can be broadly classified into two categories: the discrete detachment of large tabular icebergs (e.g. Mertz glacier tongue: Massom et al., 2015) or the spatially extensive disintegration of floating glacier tongues or ice shelves into numerous smaller icebergs (e.g. Larsen A & B ice shelves (Rott et al., 1996; Scambos et al., 2009). Observations of decadal-scale changes in glacier terminus position in both the Antarctic Peninsula and East Antarctica have suggested that despite some degree of stochasticity, iceberg calving and glacier advance/retreat is likely driven by external climatic forcing (Cook et al., 2005; Miles et al., 2013). However, despite some well-documented ice shelf collapses (Scambos et al., 2003; Banwell et al., 2013) and major individual calving events (Masson et al., 2015) there is a paucity of data on the nature and timing of calving from glaciers in Antarctica (e.g. compared to Greenland: Moon and Joughin, 2008; Carr et al., 2013), and particularly in East Antarctica.

Following recent work that highlighted the potential vulnerability of the East Antarctic Ice Sheet in Wilkes Land to ocean-climate forcing and marine ice sheet instability (Greenbaum et al., 2015; Aitken et al., 2016; Miles et al., 2013; 2016), we analyse the recent calving activity of six outlet glaciers in the Porpoise Bay region using monthly satellite imagery between November 2002 and March 2012. In addition, we also observe the start of a large calving event in 2016. We then turn our attention to investigating the drivers behind the observed calving dynamics.

## 2. Study area

Porpoise Bay (-76°S, 128°E) is situated in Wilkes Land, East Antarctica, approximately 300 km east of Moscow University Ice Shelf and 550 km east of Totten glacier (Fig. 1). This area was selected for study because it occupies a central position in Wilkes Land, which is thought to have experienced mass loss over the past decade (King et al., 2012; Sasgen et al., 2013; McMillan et al., 2014), and which is the only region of East Antarctica where the majority of marine-terminating outlet glaciers have experienced recent (2000-2012) retreat (Miles et al., 2016). This is particularly concerning because Wilkes Land overlies the Aurora subglacial basin and, due its reverse bed slope and deep troughs (Young et al., 2011), it may have been susceptible to unstable grounding line retreat in the past (Cook et al., 2014), and could make significant contributions to global sea level in the future (DeConto and Pollard, 2016). However, despite some analysis on glacier terminus position on a decadal timescales (Frezzotti and Polizzi, 2002; Miles et al., 2013; 2016), there has yet to be any investigation of inter-annual and sub-annual changes in terminus position and calving activity in the region.

Porpoise Bay is 150 km wide and is typically filled with land-fast multi-year sea-ice (Fraser et al., 2012). In total, six glaciers were analysed, with glacier velocities (from Rignot et al., 2011) ranging from ~440 m yr$^{-1}$ (Sandford Glacier) to ~2000 m yr$^{-1}$ (Frost Glacier). Recent studies have suggested that the largest (by width) glacier feeding into the bay - Holmes Glacier - has been thinning over the past decade (Pritchard et al., 2009;  McMillan et al., 2014).

## 3. Methods

### 3.1 Satellite imagery and terminus position change

Glacier terminus positions were mapped at approximately monthly intervals between November 2002 and March 2012, using Envisat Advanced Synthetic Aperture Radar (ASAR) Wide Swath Mode (WSM) imagery across six glaciers, which were identified from the Rignot et al. (2011b) ice velocity dataset (Fig.1). Additional sub-monthly imagery between December 2006 and April 2007 were used to gain a higher temporal resolution following the identification of a major calving event around that time. During the preparation for this manuscript we also observed the start of another large calving event, which we observed with Sentintel-1 imagery (Table 1).

Approximately 65% of all glacier frontal measurements were made using an automated
mapping method. This was achieved by automatically classifying glacier tongues and sea-ice
into polygons based on their pixel values, with the boundary between the two taken as the
terminus positon. The threshold between glacial ice and sea-ice was calculated automatically
based on the image pixel statistics, whereby sea-ice appears much darker than the glacial ice.
In images where the automated method was unsuccessful, terminus position was mapped
manually. The majority of these manual measurements were undertaken in the austral
summer (December – February) when automated classification was especially problematic
due to the high variability in backscatter on glacier tongues as a result of surface melt.
Following the mapping of the glacier termini, length changes were calculated using the box
method (Moon and Joughin, 2008). This method calculates the glacier area change between
each time step divided by the width of the glacier, to give an estimation of glacier length
change. The width of glacier was obtained by a reference box which approximately delineates
the sides of the glacier.
Given the nature of the heavily fractured glacier fronts and the moderate resolution of Envisat
ASAR WSM imagery (80 m) it was sometimes difficult to establish if individual or blocks of
icebergs were attached to the glacier tongue. As a result, there are errors in precisely
determining terminus change on a monthly time-scale (~± 500 m). However, because our
focus is on major calving events, absolute terminus position is less important than the
identification of major episodes of calving activity. Indeed, because estimations of terminus
position were made at approximately monthly intervals, calving events were easily
distinguished because the following month's estimation of terminus position would clearly
show the glacier terminus in a retreated position. In addition, each image was also checked
visually to make sure no small calving events were missed (i.e. as indicated by the presence
of icebergs proximal to the glacier tongue).
**3.2 Sea-ice**
Sea-ice concentrations in Porpoise Bay were calculated using mean monthly Bootstrap sea-
ice concentrations derived from the Nimbus-7 satellite and the Defence Meteorological
Satellite Program (DMSP) satellites which offers near complete coverage between October
1978 and December 2014 (Comiso, 2014; http://dx.doi.org/10.5067/J6JQLS9EJ5HU). To
extend the sea-ice record, we also use mean monthly Nimbus-5 Electrically Scanning
Microwave Radiometer (ESMR) derived sea-ice concentrations (Parkinson et al., 2004;
https://nsidc.org/data/docs/daac/nsidc0009_esmr_seaice.gd.html), which offer coverage
between December 1972 and March 1977. However, from March to May 1973, August 1973,
April 1974 and June to August 1975, mean monthly sea-ice concentrations were not
available. Sea-ice concentrations were extracted from 18 grid cells, covering 11,250 km$^2$ that
extended across Porpoise Bay, but not into the extended area beyond the limits of the bay
(Fig. 1). Grid cells which were considered likely to be filled with glacial ice were excluded.
Pack ice concentrations were also extracted from a 250 x 150 km polygon adjacent to
Porpoise Bay. The dataset has a spatial resolution of 25 km and monthly sea-ice
concentration anomalies were calculated from the 1972-2016 monthly mean.
Daily sea-ice concentrations derived from the Artist Sea-Ice (ASI) algorithm from Advanced
Microwave Scanning Radiometer - EOS (AMSR-E) data (Spreen et al., 2008) were used to
calculate daily sea-ice concentration anomalies during the January 2007 sea-ice break-up
(http://icdc.zmaw.de/1/daten/cryosphere/seaiceconcentration-asi-amsre.html). This dataset
was used because it provides a higher spatial resolution (6.25 km) compared to those
available using Bootstrap derived concentrations (25 km). This is important because it
provides a more accurate representation of when sea-ice break-up was initiated and, due to its
much higher spatial resolution, it provides data from much closer to the glacier termini (see
Fig.1).

### 3.3 RACMO

We used the Regional Atmospheric Climate Model (RACMO) V2.3 (van Wessem et al., 2014)
to simulate daily surface melt fluxes in the study area between 1979 and 2015 at a 27 km
spatial resolution. The melt values were extracted from floating glacier tongues in Porpoise
Bay because the model masks out sea-ice, equating to seven grid points. The absolute surface
melt values are likely to be different on glacial ice, compared to the sea-ice, but the relative
magnitude of melt is likely to be similar temporally.

### 3.4 ERA-interim

In the absence of weather stations in the vicinity of Porpoise Bay we use the 0.25° ERA-
interim reanalysis dataset (http://apps.ecmwf.int/datasets/data/interim-full-
moda/levtype=sfc/) to calculate mean monthly wind field and sea surface temperature (SST)
anomalies, with respect to the 1979-2015 monthly mean. Wind field anomalies were
calculated by using the mean monthly 10 m zonal (U) and meridional (V) wind components.
We also used the daily 10 m zonal (U) and meridional (V) components to simulate wind field
vectors in Porpoise Bay on January 11[th] 2007 and March 19[th] 2016 which are the estimated
dates of sea-ice break-up.

## 4. Results

### 4.1 Terminus position change

Analysis of glacier terminus position change of six glaciers in Porpoise Bay between
November 2002 and March 2012 reveals three broad patterns of glacier change (Fig. 2). The
first pattern is shown by Holmes (West) glacier, which advances a total of ~13 km throughout
the observation period, with no evidence of any major iceberg calving that resulted in
substantial retreat of the terminus beyond the measurement error (+/- 500 m). The second is
shown by Sandford Glacier tongue, which advanced ~1.5 km into the ocean between
November 2002 and April 2006, before its floating tongue broke away in May 2006. A further
smaller calving event was observed in January 2009. Overall, by the end of the study period,
its terminus had retreated around 1 km from its position in November 2002. The third pattern is
shown by Frost Glacier, Glacier 1, Glacier 2 and Holmes (East) glaciers, which all advanced
between November 2002 and January 2007, albeit with a small calving event in Frost glacier in
May 2006. However, between January and April 2007, Frost Glacier, Glacier 1, Glacier 2 and
Holmes (East) glaciers all underwent a large near-simultaneous calving event. This led to
1,300 km$^2$ of ice being removed from glaciers in Porpoise Bay, although we also note the
disintegration of a major tongue from an unnamed glacier further west, which contributed a
further 1,600 km$^2$. Thus, in a little over three months, a total of 2,900 km$^2$ of ice was removed
from glacier tongues in the study area (Fig. 3). Following this calving event, the fronts of these
glaciers stabilised and began advancing at a steady rate until the end of the study period
(March, 2012) (Fig. 2), with the exception of Frost glacier which underwent a small calving
event in April 2010.

### 4.2 Evolution of the 2007 calving event

A series of eight sub-monthly images between December 11[th] 2006 and April 8[th] 2007 show
the evolution of the 2007 calving event (Fig. 4). Between December 11[th] 2006 and January 2[nd]
2007, the land-fast sea-ice edge retreats past Sandford glacier to the edge of Frost glacier and
there is some evidence of sea-ice fracturing in front of the terminus of Glacier 2 (Fig. 4b).
From January 2[nd] to January 9[th] a small section (~40 km$^2$) of calved ice broke away from Frost
glacier, approximately in line with the retreat edge of land-fast sea-ice (Fig. 4c). By January
25th, significant fracturing in the land-fast sea-ice had developed, and detached icebergs from
Frost, Glacier 1, Glacier 2 and Holmes East glaciers begin to breakaway (Fig. 4d). This process
of rapid sea-ice breakup in the east section of the bay and the disintegration of sections of Frost
glacier, Glacier 1, Glacier 2 and Holmes East glaciers continues up to March 10th 2007 (Fig.
4g). In contrast, the west section of Porpoise Bay remains covered in sea-ice in front of
Holmes west glacier, which does not calve throughout this event. By April 8th, the calving
event had ended with a large number of calved icebergs now occupying the bay (Fig. 4h).
**4.3 2016 calving event**
During the preparation of this manuscript satellite observations of Porpoise Bay revealed that
another large near-simultaneous disintegration of glacier tongues in Porpoise Bay is currently
underway. This event was initiated on March 19th where the edge of the multi-year sea-ice
retreated to the Holmes West glacier terminus, removing multi-year sea-ice which was at
least 14 years old. By March 24th this had led to the rapid disintegration of an 800 km$^2$
section of the Holmes West glacier tongue (Fig. 5). This was the first observed calving of
Holmes (West) glacier at any stage between November 2002 and March 2016. Throughout
March and April the break-up of sea-ice continued and by May 13th it had propagated to the
terminus of Frost Glacier, resulting in the disintegration of large section of its tongue (Fig. 6).
By 24th July sea-ice had been removed from all glacier termini in Porpoise Bay at some point
during the event, resulting in a total of ~2,200 km$^2$ ice being removed from glacier tongues
(Fig. 6).
**4.4. The link between sea-ice and calving in Porpoise Bay**
Analysis of mean monthly sea-ice concentration anomalies in Porpoise Bay between
November 2002 and June 2016 (Fig. 7) reveals a major negative sea-ice anomaly occurred
between January and June 2007, where monthly sea-ice concentrations were between 35%
and 40% below average. This is the only noticeable (>20%) negative ice anomaly in Porpoise
Bay and it coincides with the major January 2007 calving event (see Fig. 4). However,
despite satellite imagery showing the break-up of sea-ice prior to the 2016 calving event (Fig.
5 and 6), in a similar manner to that in 2007 (e.g. Fig. 4), no large negative anomaly is
present in the sea-ice concertation data. This is likely to reflect the production of a large
armada of icebergs following the disintegration of Holmes (West) Glacier (e.g. Fig. 6) ,
helping promote a rapid sea-ice reformation in the vicinity of Porpoise Bay. Furthermore, we
note that the smaller calving events of Sandford and Frost glaciers all take place after sea-ice
had retreated away from the glacier terminus (Fig. 8). Indeed, throughout the study period,
there is no evidence of any calving events taking place with sea-ice proximal to glacier
termini. This suggests that glaciers in Porpoise Bay are very unlikely to calve with sea-ice
present at their termini.
**4.5. Atmospheric circulation anomalies**
Atmospheric circulation anomalies in the months preceding the January 2007 and March
2016 sea-ice break-ups reveal contrasting conditions. In the austral summer which preceded
the January 2007 break-up there were strong positive SST anomalies and atmospheric
circulation anomalies throughout December 2005 (Fig. 9a). The circulation anomaly was
reflected in a strong  easterly airflow offshore from Porpoise Bay. This is associated with a
band of cooler SSTs close to the coastline and the northward shift of the Antarctic Coastal
Current in response to the weakened westerlies (e.g. Langlais et al., 2015). A weakened zonal
flow combined with high sea surface temperatures (SST) in the south Pacific would allow the
advection of warmer maritime air into Porpoise Bay. Consistent with warmer air  are
estimates of exceptionally high melt values in Porpoise Bay during December 2005 derived
from the RACMO2.3, which contrasts with the longer-term trend of cooling (Fig. 10).
However, the December 2005 anomaly was short -ved and, by January 2006, the wind field
conditions were close to average, although SST remained slightly higher than average (Fig.
9b).
In December 2006 and January 2007, which are the months immediately before and during
the break-up of sea-ice, atmosphere conditions were close to average, with very little
deviation from mean conditions in the wind field and a small negative SST anomaly (Fig.
9c). However, on January 11[th] 2007, which is the estimated date of sea-ice break-up from
AMSR-E data, we note that there were very high winds close to Porpoise Bay (Fig. 11a).
In contrast to the months preceding the January 2007 event, we find little deviations from
average conditions prior to the March 2016 break-up event. In the austral summer which
preceded the 2016 break-up (2014/15), there was little deviation from the average wind field
and only a small increase from average SSTs (Fig 9d). In December and January 2015/16,
there was evidence for a small increase in the strength of westerly winds, and cooler SSTs in
the South Pacific (Fig. 9e). However, in February and March 2016 there was no change from
the average wind field and slightly cooler SSTs (Fig. 9f).We note, however, that there was a
low pressure system passing across Porpoise Bay on March 19[th] 2016, the estimated date of
break-up initiation (Fig. 11b).

### 4.6 Holmes (West) Glacier calving cycle

Through mapping the terminus position in all available satellite imagery (Table 1) dating
back to 1963, we are able to reconstruct large calving events on the largest glacier in Porpoise
bay, Holmes (West) (Fig. 12). On the basis that a large calving event is likely during the
largest sea-ice break-up events, we estimate the date of calving based on sea-ice
concentrations in Porpoise Bay when satellite imagery is not available. Our estimates suggest
that Holmes (West) Glacier calves at approximately the same positon in each calving cycle,
including the most recent calving event in March 2016.

### 5. Discussion

### 5.1 Sea-ice break-up and the disintegration of glacier tongues in Porpoise Bay

We report a major, near-synchronous calving event in January 2007 and a similar event that
was initiated in 2016 and resulted in ~2,900 km$^2$ and 2,200 km$^2$ of ice, respectively, being
removed from glacier tongues in the Porpoise Bay region of East Antarctica. This is
comparable to some of the largest disintegration events ever observed in Antarctica (e.g.
Larsen A in 1995, 4,200 km$^2$ and Larsen B in 2002, 3,250 km$^2$); and is the largest event to have
been observed in East Antarctica. However, this event differs from those observed on the ice
shelves of the Antarctic Peninsula, in that it may be more closely linked to a cycle of glacier
advance and retreat, as opposed to a catastrophic collapse that may be unprecedented.
Given the correspondence between the sea-ice and glacier terminus changes, we suggest that
these disintegration events were driven by the break-up of the multi-year land-fast sea-ice
which usually occupies Porpoise Bay and the subsequent loss of buttressing of the glacier
termini. A somewhat similar mechanism has been widely documented in Greenland, where the
dynamics of sea-ice melange in proglacial fjords has been linked to inter-annual variations in
glacier terminus position (Amundson et al., 2010; Carr et al., 2013; Todd and Christoffersen,
2014; Cassotto et al., 2015). Additionally, the mechanical coupling between thick multi-year
landfast sea ice and glacier tongues may have acted to stabilize and delay the calving of the
Mertz glacier tongue (Massom et al., 2010) and Brunt/Stancomb-Wills Ice Shelf system
(Khazendar et al., 2009). However, this is the first observational evidence directly linking
multi-year landfast sea-ice break-up to the large scale and rapid disintegration of glacier

tongues. This is important because landfast sea-ice is highly sensitive to climate (Heil, 2006; Mahoney et al., 2007) and, if future changes in climate were to result in a change to the persistence and/or stability of the landfast ice in Porpoise Bay, it may result in detrimental effects on glacier tongue stability. An important question, therefore, is: what process(es) cause sea-ice break-up?

**5.2 What caused the January 2007 and March 2016 sea-ice break-ups?**

The majority of sea-ice in Porpoise Bay is multi-year sea-ice (Fraser et al., 2012), and it is likely that various climatic processes operating over different timescales contributed to the January 2007 sea-ice break-up event. Although there are no long-term observations of multi-year sea-ice thickness in Porpoise Bay, observations and models of the annual cycle of multi-year sea-ice in other regions of East Antarctica suggest that multi-year sea-ice thickens seasonally and thins each year (Lei et al., 2010; Sugimoto et al., 2016; Yang et al., 2016). Therefore, the relative strength, stability and thickness of multi-year sea ice at a given time period is driven not only by synoptic conditions in the short term (days/weeks), but also by climatic conditions in the preceding years.

In the austral summer (2005/06) which preceded the break-up event in January 2007, there was a strong easterly airflow anomaly throughout December 2005 directly adjacent to Porpoise Bay (Fig. 9a). This anomaly represents the weakening of the band of westerly winds which encircle Antarctica, and is reflected in an exceptionally negative Southern Annular Mode (SAM) index in December 2005 (Marshal, 2003). This contrasts with the long-term trend for a positive SAM index (Marshal, 2007; Miles et al., 2013). A weaker band of westerly winds combined with anomalously high SST in the Southern Pacific (Fig. 9a) would allow a greater advection of warmer maritime air towards Porpoise Bay. Indeed, RACMO2.3 derived surface melt estimates place December 2005 as the second highest mean melt month (1979-2015) on the modelled output in Porpoise Bay (Fig. 10). To place this month into perspective, we note that it would rank above the average melt values of all Decembers and Januarys since 2000 on the remnants of Larsen B ice shelf. Comparing MODIS satellite imagery from before and after December 2005 reveals the development of significant fracturing in the multi-year sea-ice (Fig 13a, b). These same fractures remain visible prior to the break-out event in January 2007 and, when the multi-year sea-ice begins to break-up, it ruptures along these pre-existing weaknesses (Fig. 13c). As such, this strongly suggests that the atmospheric circulation anomalies of

December 2005 played an important role in the January 2007 multi-year sea-ice break-up and
near-simultaneous calving event.
The break-up of landfast sea-ice has been linked to dynamic wind events and ocean swell
(Heil, 2006; Ushio, 2006; Fraser et al., 2012). Thus, it is possible that the wind anomalies in
December 2005 may have been important in initiating the fractures observed in the sea-ice in
Porpoise Bay, through changing the direction and/or intensity of oceanic swell. However, this
mechanism is thought to be at its most potent during anonymously low pack-ice concentrations
because pack-ice can act as a buffer to any oceanic swell (Langhorne et al., 2001; Heil, 2006;
Fraser, 2012). That said, we note that pack-ice concentrations offshore of Porpoise Bay were
around average during December 2005 (Fig. 7). This may suggest that there are other
mechanisms that were important in the weakening of the multi-year sea-ice in Porpoise Bay in
December 2005.
In the Arctic, sea-ice melt ponding along pre-existing weaknesses has been widely reported to
precede sea-ice break-up (Ehn et al., 2011; Petrich et al., 2012; Landy et al., 2014; Schroder et
al., 2014; Arntsen et al., 2015). Despite its importance in the Arctic, it has yet to be considered
as a possible factor in landfast sea-ice break-up in coastal Antarctica. As a consequence of the
high melt throughout December 2005, the growth of sea-ice surface ponding would be
expected, in addition to surface thinning of the sea-ice. High-resolution cloud free optical
satellite coverage of Porpoise Bay throughout December 2005 is limited, but ASTER imagery
in the vicinity of Frost Glacier on the 4[th] and 31[st] December 2005 shows surface melt features
and the development of fractures throughout the month (Fig. 13d,e), similar to those observed
elsewhere in East Antarctica (Kingslake et al., 2015; Langley et al., 2016).High-resolution
imagery from 16[th] January 2006 (via GoogleEarth) shows the development of melt ponds on
the sea-ice surface (Fig. 13f). Therefore, it is possible that surface melt had some impact on the
fracturing of landfast sea-ice in Porpoise Bay. This may have caused hydro-fracturing of pre-
existing depressions in the landfast ice or surface thinning may have made it more vulnerable
to fracturing through ocean swell or internal stresses. Additionally, the subsequent refreezing
of some melt ponds may temporally inhibit basal ice growth, potentially weakening the multi-
year sea-ice and presdiposing it to future break-up (Flocco et al., 2015). It is important to note
that the atmospheric circulation anomalies which favoured the development of fractures in the
multi-year sea-ice in December 2005 were short-lived. By January 2006, atmospheric
conditions had returned close to average (Fig. 9b) and remained so until the austral winter,
where sea-ice break-up is less likely. This may explain the lag between the onset of sea-ice
fracturing in December 2005 and its eventual break-up in the following summer (January
347 2007).

Consistent with the notion that the multi-year sea-ice was already in a weakened state prior to
its break-up in 2007, is that the break-up occurred in January, several weeks before the likely
annual minimums in multi-year sea-ice thickness (Yang et al., 2016; Lei et al., 2010) and
landfast ice extent (Fraser et al., 2012). Additionally, atmospheric circulation anomalies
indicate little deviation from average conditions in the immediate months preceding break-up
(Fig. 9b, c), suggesting that atmospheric conditions were favourable for sea-ice stability.
Despite this, a synoptic event is still likely required to force the break-up in January 2007.
Daily sea-ice concentrations in Porpoise Bay in January 2007 January show a sharp decrease in
sea-ice concentrations after $12^{th}$ January, representing the onset of sea-ice break-out (Fig 14).
This is preceded by a strong melt event recorded by the RACMO2.3 model, centred on January
$11^{th}$, which may represent a low pressure system. Indeed, ERA-interim estimates of the wind
field suggest strong south-easterly winds in the vicinity of Porpoise Bay (Fig 11 a). Unlike in
December 2005, pack ice concentrations offshore of Porpoise Bay were anonymously low
(Fig. 7). Therefore, with less pack ice buttressing, it is possible that the melt event, high winds
and associated ocean swell may have initiated the break-up of the already weakened multi-year
sea-ice in Porpoise Bay.
In contrast to January 2007, we find no link between atmospheric circulation anomalies and
the March 2016 sea-ice break-up. In the preceding months to the March 2016 break-up, wind
and SST anomalies indicate conditions close to average conditions favouring sea-ice stability
(Fig. 9 d, e, f). This suggests another process was important in driving the March 2016 sea-
ice break-up. A key difference between the 2007 and 2016 event is that the largest glacier in
the bay, Holmes (West), only calved in the 2016 event. Analysis of its calving cycle (Fig. 12)
indicates that it calves at roughly the same position in each cycle and that its relative position
in early 2016 suggests that calving was 'overdue' (Fig. 12). This indicates that the calving
cycle of Holmes (West) Glacier is not necessarily been driven by atmospheric circulation
anomalies. Instead, we suggest that as Holmes (West) Glacier advances, it slowly pushes the
multi-year sea-ice attached to its terminus further towards the open ocean to the point where
the sea-ice attached to the glacier tongue becomes more unstable. This could be influenced by
local bathymetry and oceanic circulation, but no observations are available. However, once
the multi-year sea-ice reaches an unstable state, break-up is still likely to be forced by a
synoptic event. This is consistent with our observations, where ERA-interim derived wind
fields show the presence of a low pressure system close to Porpoise Bay on the estimated date
of sea-ice break-up in March 2016 (Fig. 11 b). Whilst we suggest that the March 2016 sea-ice
break-up and subsequent calving of Holmes (West) is currently part of a predictable cycle,
we note that this could be vulnerable to change if any future changes in climate alter the
persistence and/or strength of the multi-year sea-ice, which is usually attached to the glacier
terminus.
**6.  Conclusion**
We identify two large near-simultaneous calving events in January 2007 and March 2016
which were driven by the break-up of the multi-year landfast sea-ice which usually occupies
the bay. This provides a previously unreported mechanism for the rapid disintegration of
floating glacier tongues in East Antarctica, adding to the growing body of research linking
glacier tongue stability to the mechanical coupling of landfast ice (e.g. Khazander et al.,
2009; Massom et al., 2010). Our results suggest that multi-year sea-ice break-ups in 2007 and
2016 in Porpoise Bay were driven by different mechanisms. We link the 2007 event to
atmospheric circulation anomalies in December 2005 weakening multi-year sea-ice through a
combination of surface melt and a change in wind direction, prior to its eventual break-up in
2007. This is in contrast to the March 2016 event, which we suggest is part of a longer-term
cycle based on the terminus position of Holmes (West) Glacier that was able to advance and
push sea-ice out of the bay. The link between sea-ice break-up and major calving of glacier
tongues is especially important because it suggests predictions of future warming (DeConto
and Pollard, 2016) suggests that multi-year landfast ice may become less persistent.
Therefore, the glacier tongues which depend on landfast ice for stability may become less
stable in the future. In a wider context, our results also highlight the complex nature of the
mechanisms which drive glacier calving positon in Antarctica. Whilst regional trends in
terminus positon can be driven by ocean-climate-sea-ice interaction (e.g. Miles et al., 2013;
2016), individual glaciers and individual calving events have the potential respond differently
to similar climatic forcing.

**Acknowledgements:** We thank the ESA for providing Envisat ASAR WSM data (Project ID:
16713) and Sentinel data. Landsat imagery was provided free of charge by the U.S. Geological
Survey Earth Resources Observation Science Centre. We thank M. van den Broeke for
providing data and assisting with RACMO. B.W.J.M was funded by a Durham University

Doctoral Scholarship program. S.S.R.J. was supported by Natural Environment Research Council Fellowship NE/J018333/1. We would like to thank Allen Pope and Ted Scambos for reviewing the manuscript, along with the editor, Rob Bingham, for providing constructive comments which led to its improvement of this manuscript.

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

**Table 1: Satellite imagery used in the study**

| Satellite | Date of Imagery |
|---|---|
| ARGON | October 1963 (Kim et al., 2007) |
| Envisat ASAR WSM | August 2002, November 2002 to March 2012 (monthly) |
| Landsat | January 1973; February 1991 |
| MODIS | January 2001; December/January 2005/6; March 2016 |
| RADARSAT | September 1997 (Liu and Jezek, 2004) |
| Sentinel-1 | February-July, 2016 |



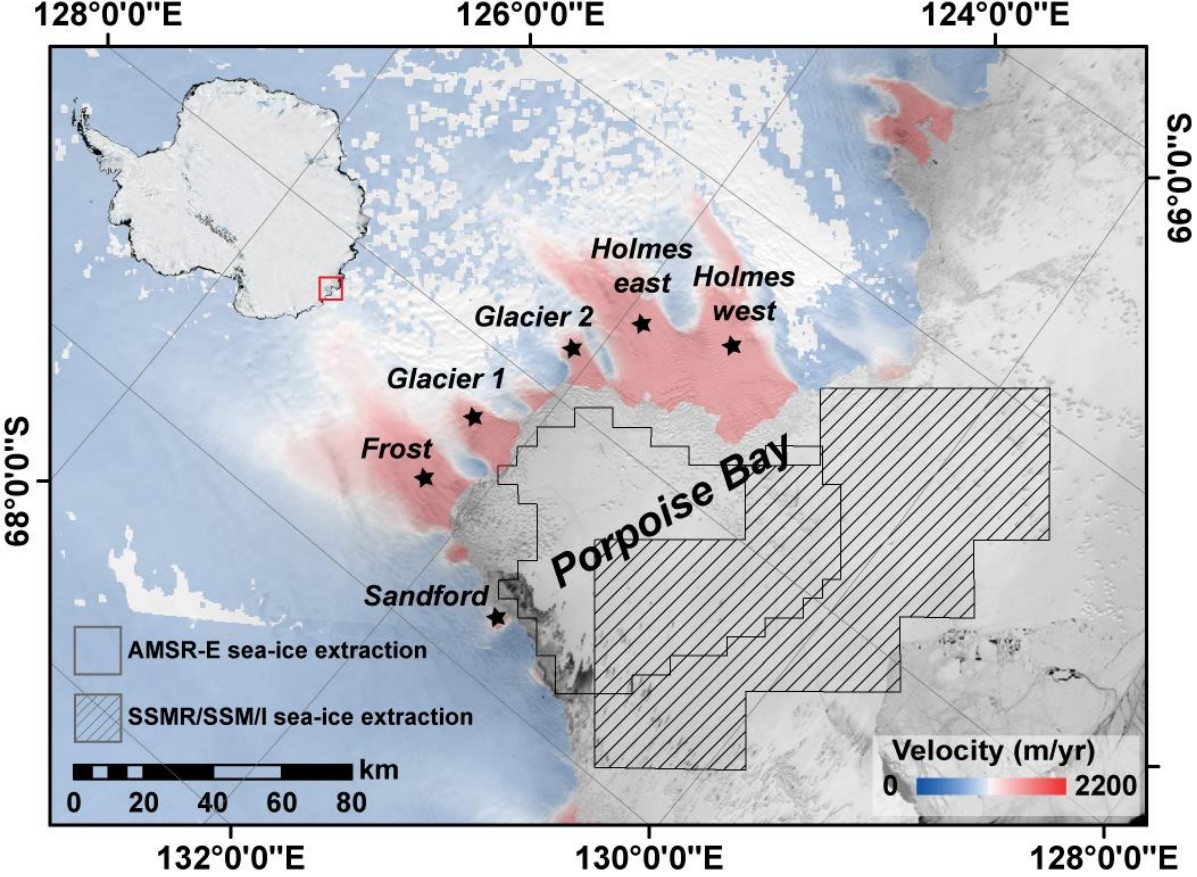


**Figure 1:** MODIS image of Porpoise Bay, with glacier velocities overlain (Rignot et al., 2011). The hatched polygon represents the region where long-term 25 km resolution SMMR/SSM/I sea-ice concentrations were extracted. The non-hatched polygon represents the region where the higher resolution (6.25 km) AMSR-E sea-ice concentrations were extracted.

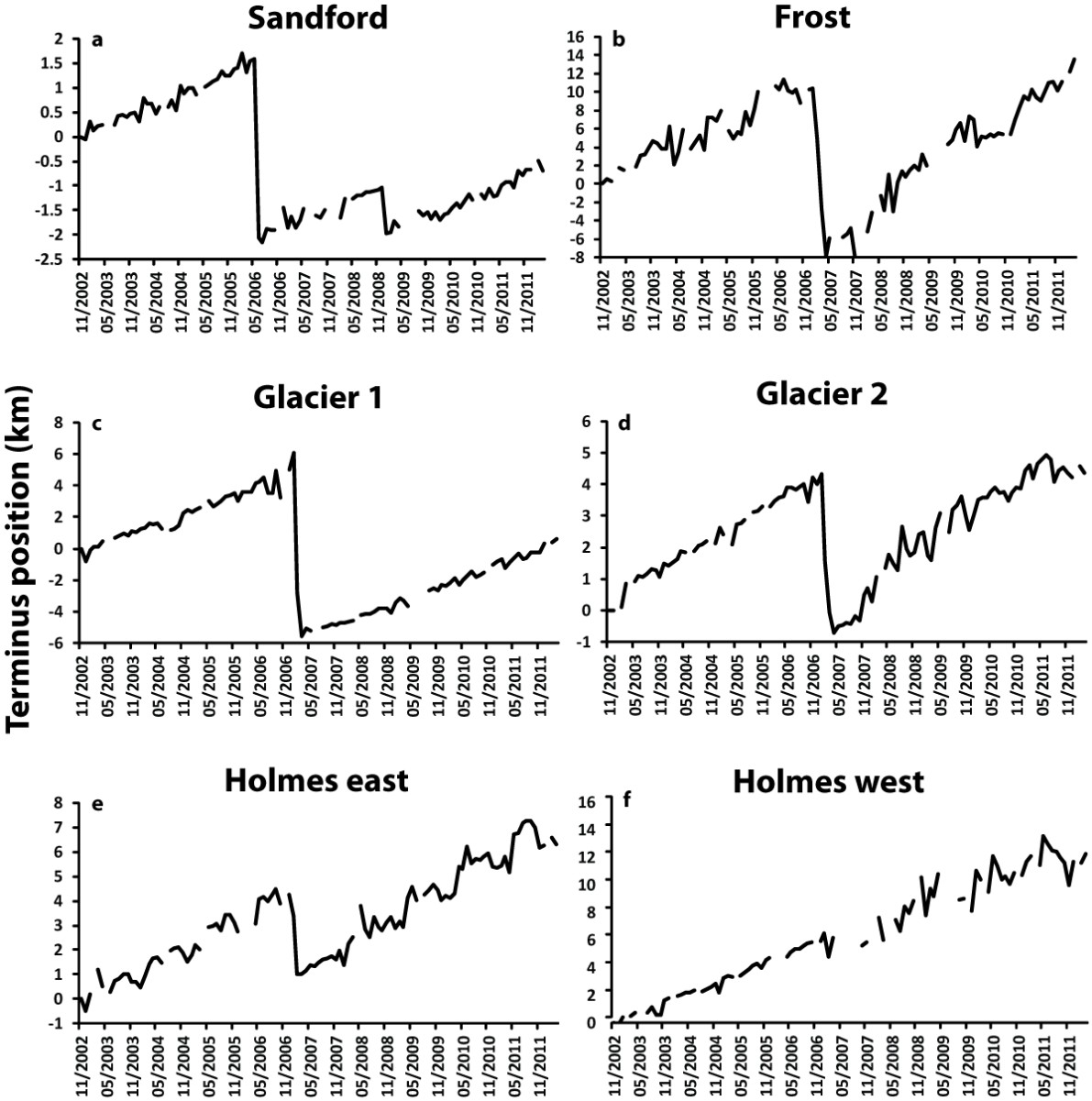

Figure 2: Terminus position change of six glaciers in porpoise Bay between November 2002 and March 2012. Note the major calving event in January 2007 for 5 of the glaciers. Terminus position measurements are subject to +/- 500 m. Note the different scales on y-axis.

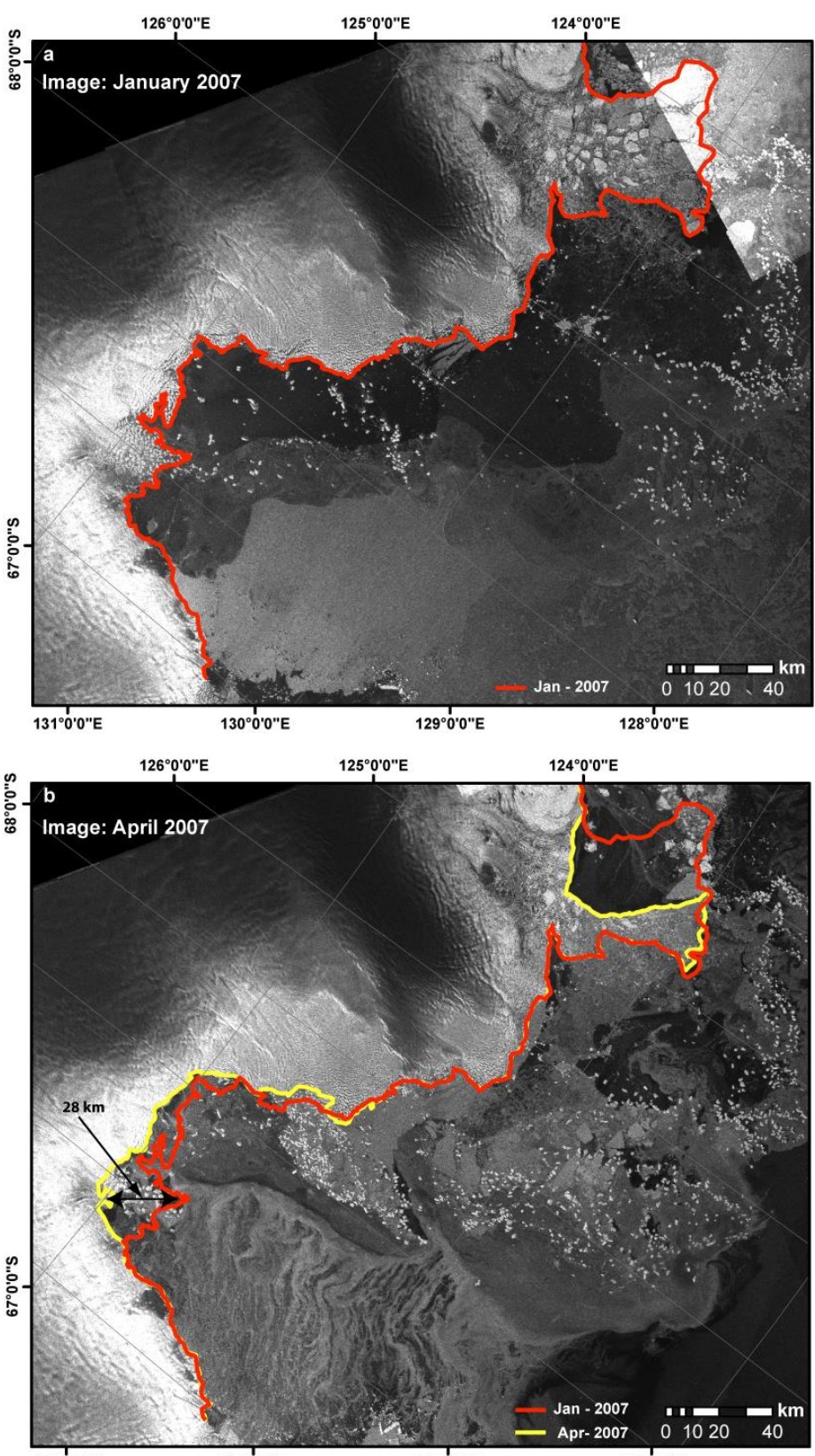


**Figure 3:** Envisat ASAR WSM imagery in January 2007 **a)** and April 2007 **b)**, which are immediately prior to and after a near-simultaneous calving event in Porpoise Bay. Red line shows terminus positions in January 2007 and yellow line shows the positions in April 2007.

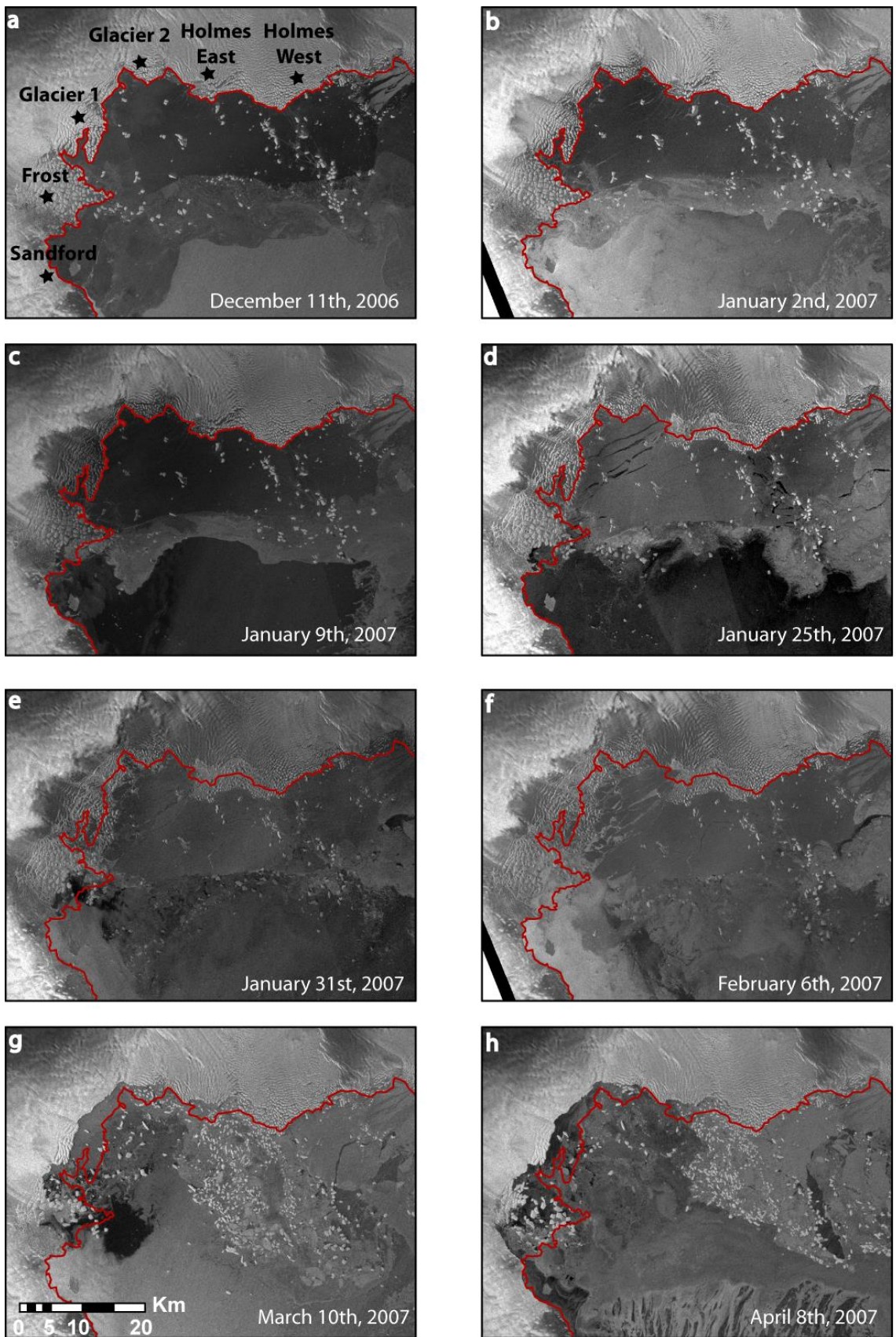


**Figure 4:** Envisat ASAR WSM imagery showing the evolution of the 2007 calving event.
Red line shows the terminus positions from December 11[th] 2006 on all panels.

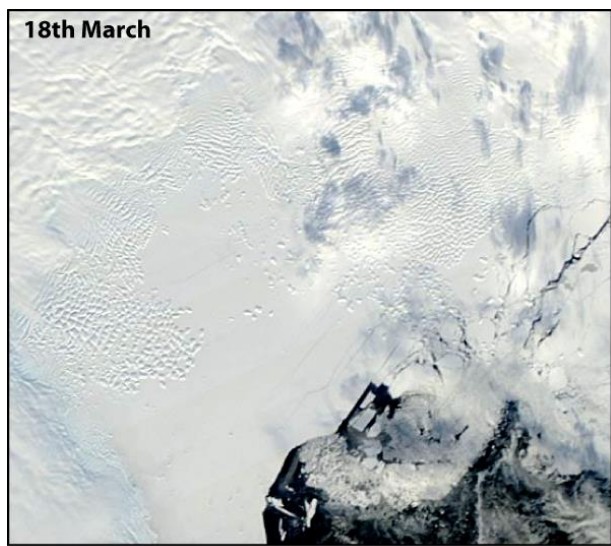

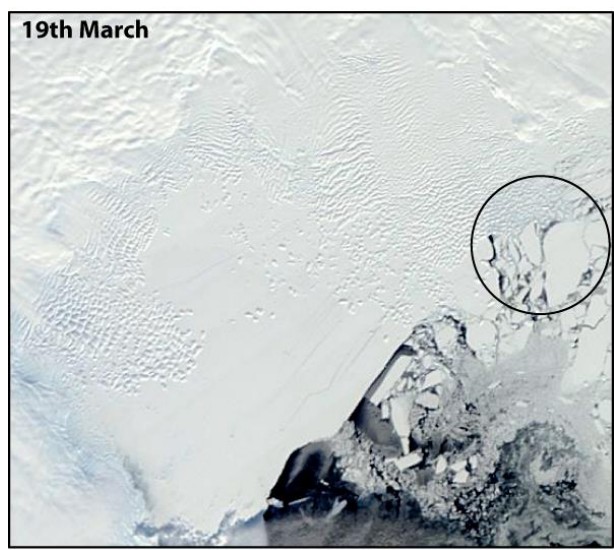

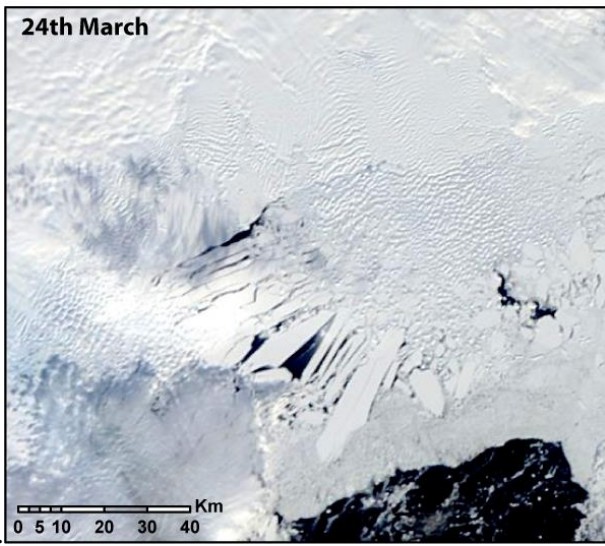

619  .

**Figure 5:** MODIS imagery showing the initial stages of disintegration of Holmes (West)
Glacier in March 2016. On March 19[th] a large section of sea-ice breaks away from the
terminus (circled), initiating the rapid disintegration process. By the 24[th] March an 800 km$^2$
section of Holmes (west) glacier tongue had disintegrated.

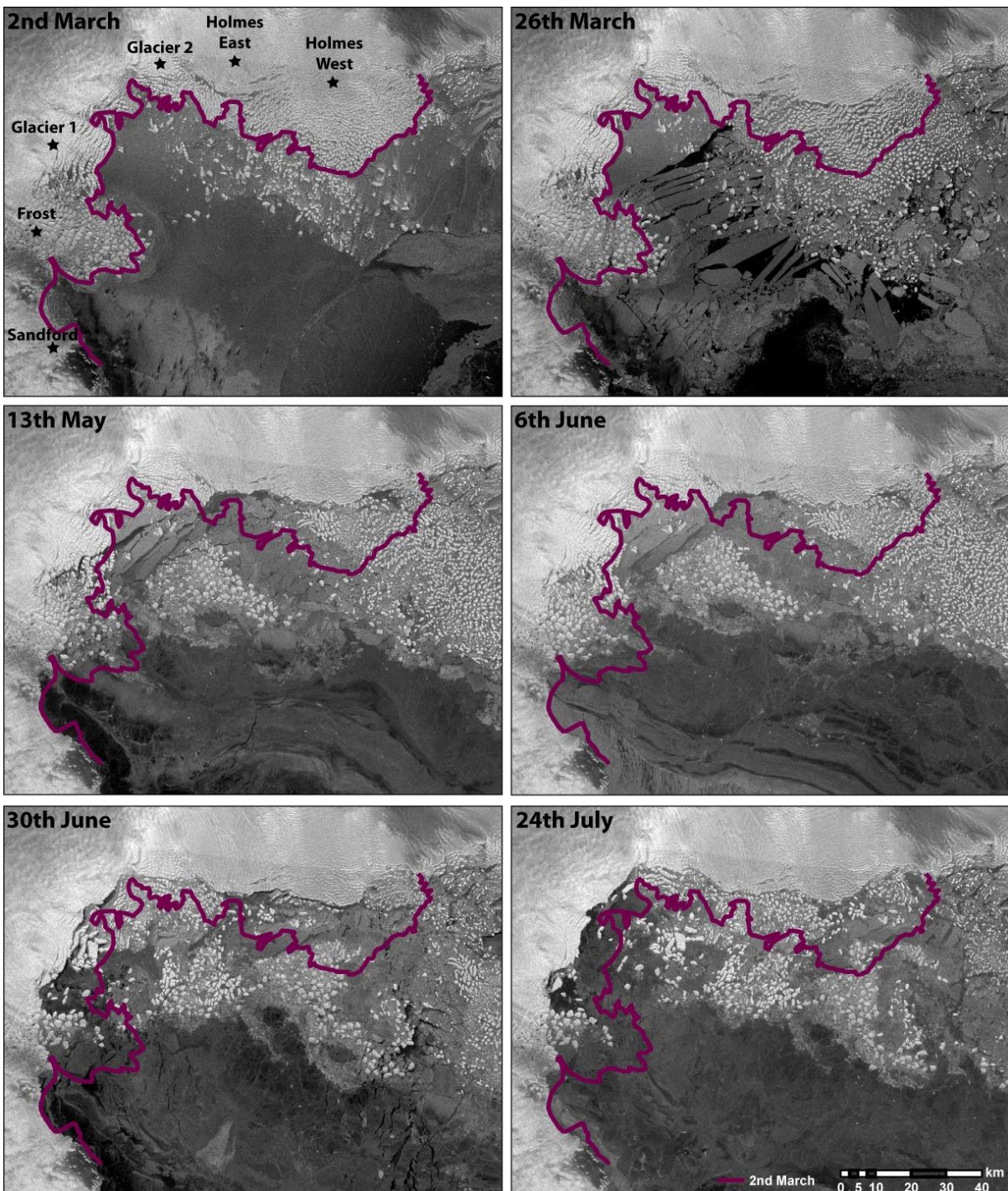


**Figure 6:** Sentinel-1 imagery showing the evolution of the 2016 calving event. Purple line shows the terminus position from 2$^{nd}$ March on all panels.

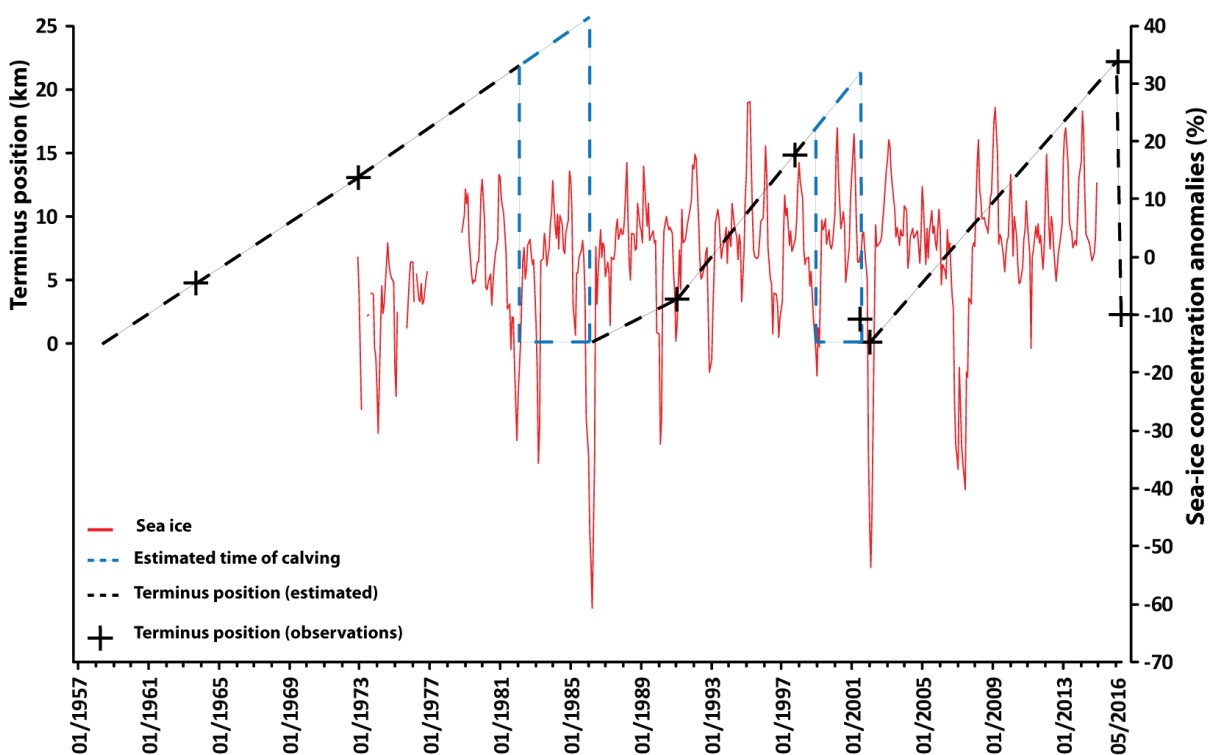


**Fig 7:** Mean monthly sea-ice concentration anomalies from November 2002 to June 2016.
The red line indicates sea-ice concentration anomalies in Porpoise Bay and the blue line
indicates pack ice concentration anomalies.

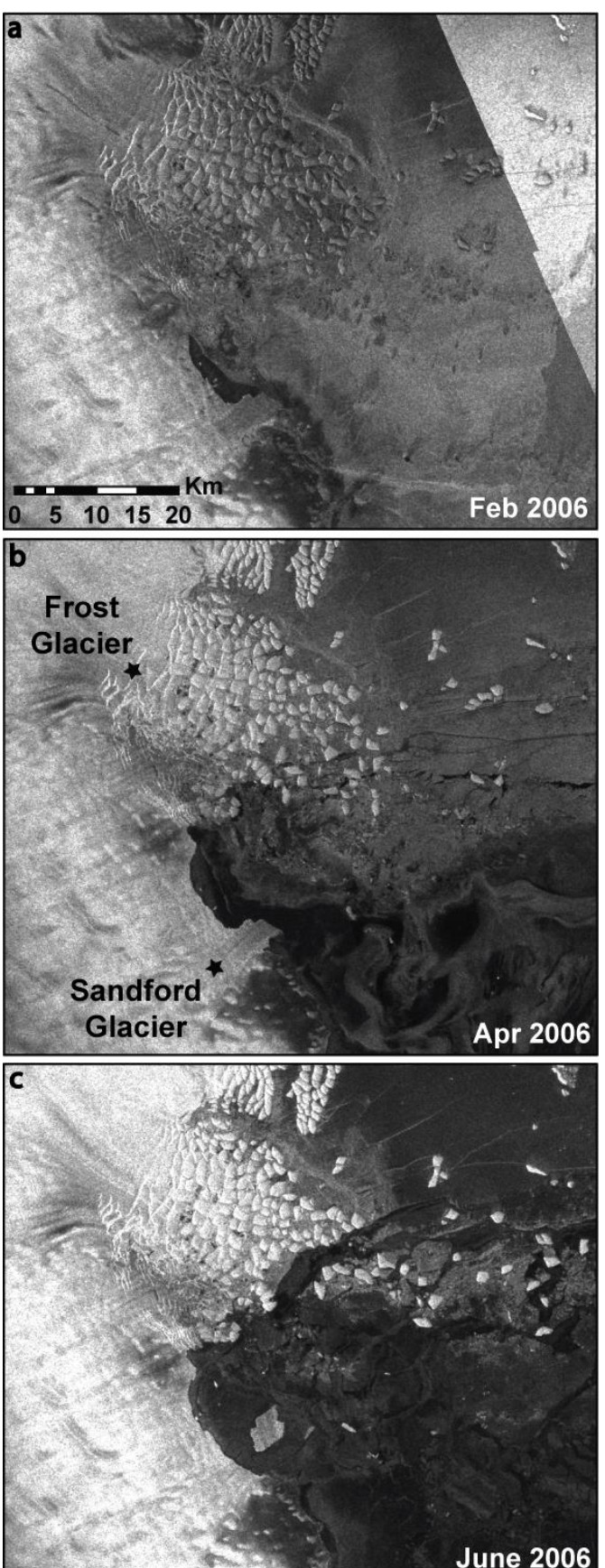


**Figure 8:** Time series of Frost and Sanford Glaciers calving showing that sea-ice clears prior
to calving and dispersal of icebergs.

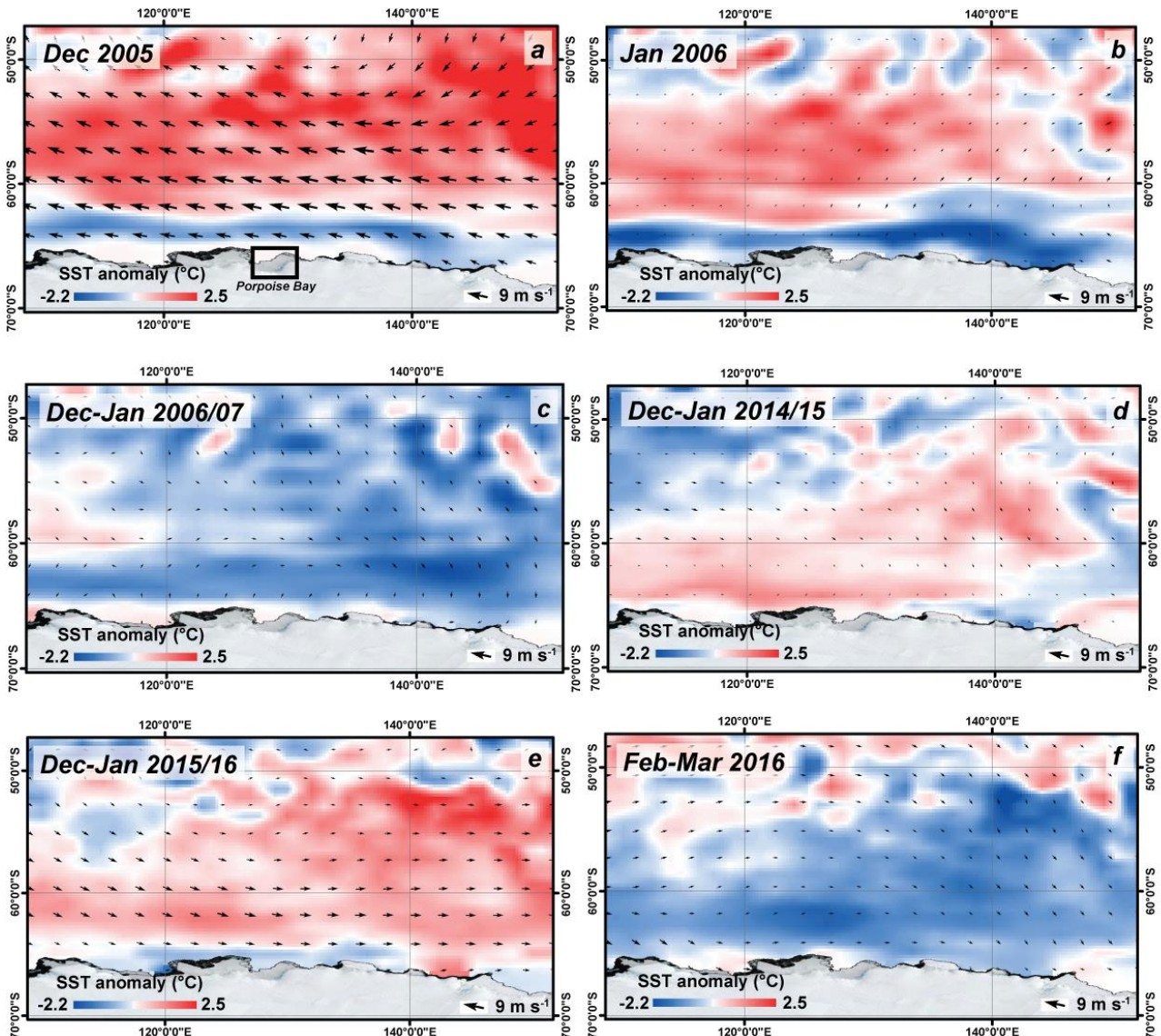

**Figure 9:** Mean monthly ERA-interim derived wind field and sea surface temperature anomalies in the months preceding the 2007 and 2016 sea-ice break-ups. **a)** December 2005 **b)** January 2006 **c)** Mean December and January 2006/07 **d)** Mean December and January 2014/15 **e)** Mean December and January 2015/16 **f)** Mean February and March 2016.

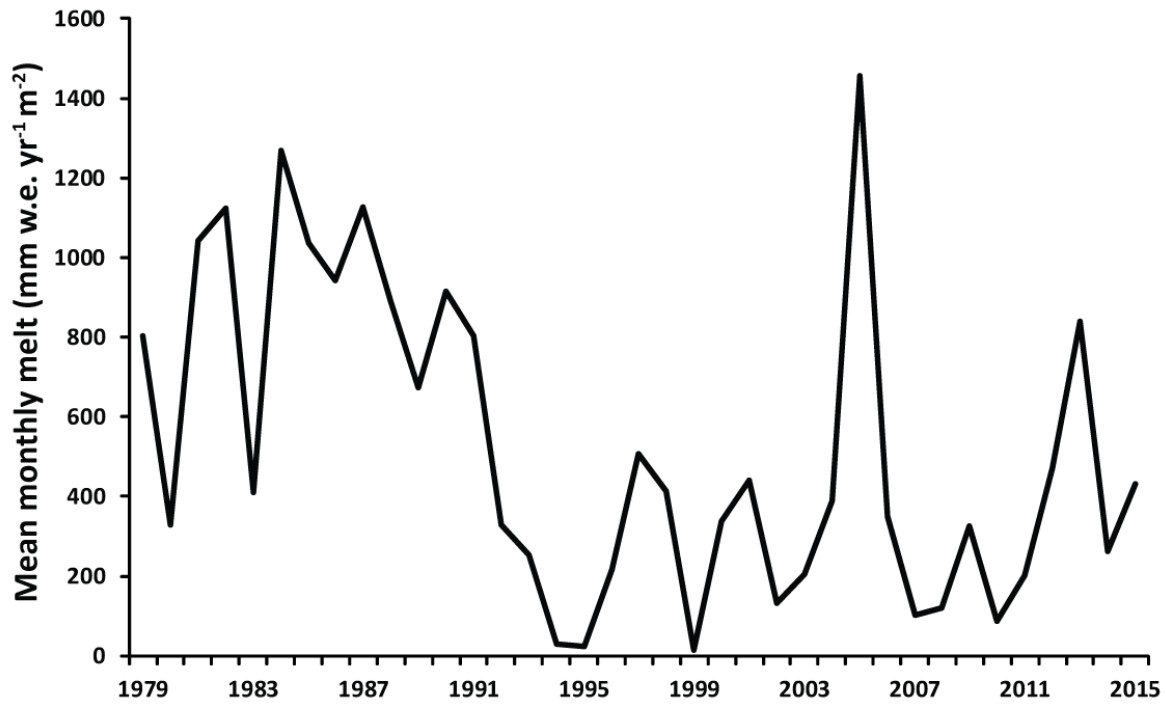


**Figure 10:** Mean RACMO2.3 derived December melt 1979-2015 in Porpoise Bay.














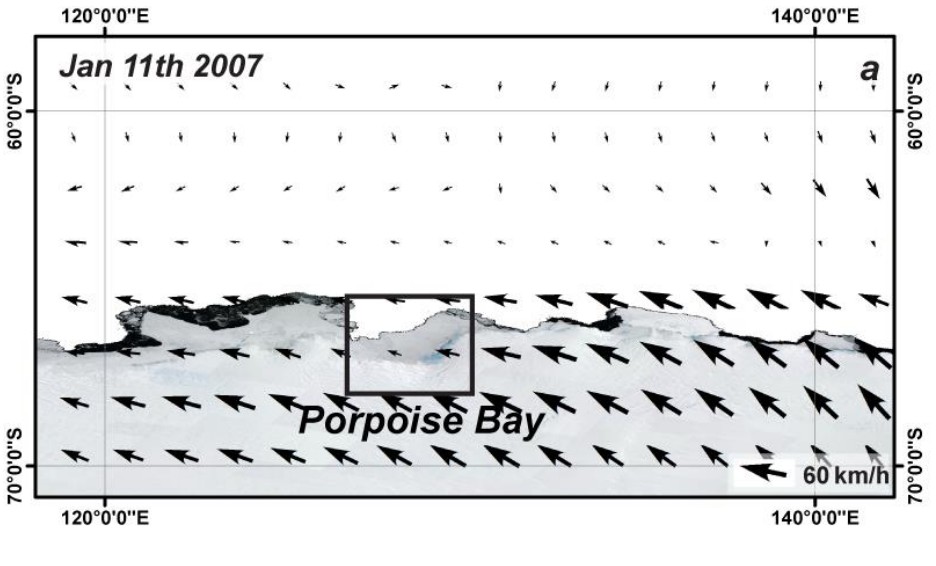

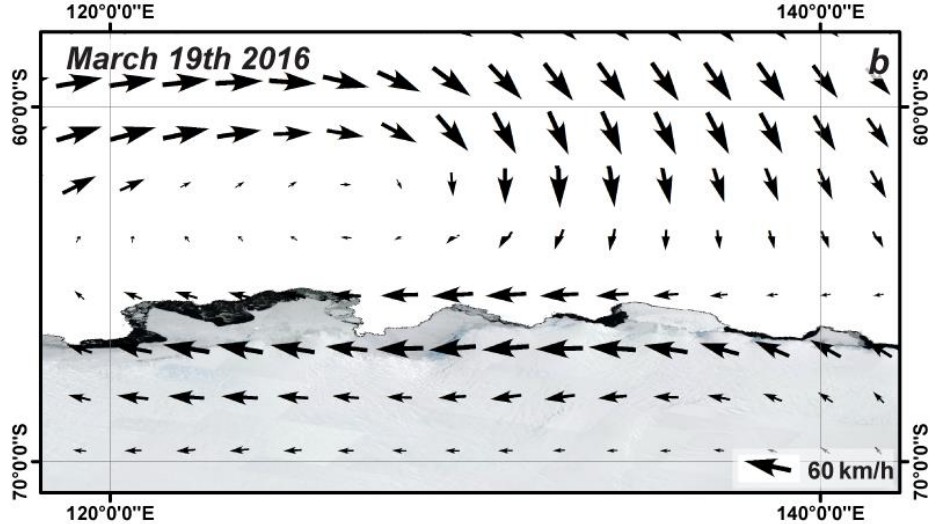

**Figure 11:** ERA-interim derived wind fields for the estimated dates of sea-ice break-up. **a)** January 11[th] 2007 and **b)** March 19[th] 2016.

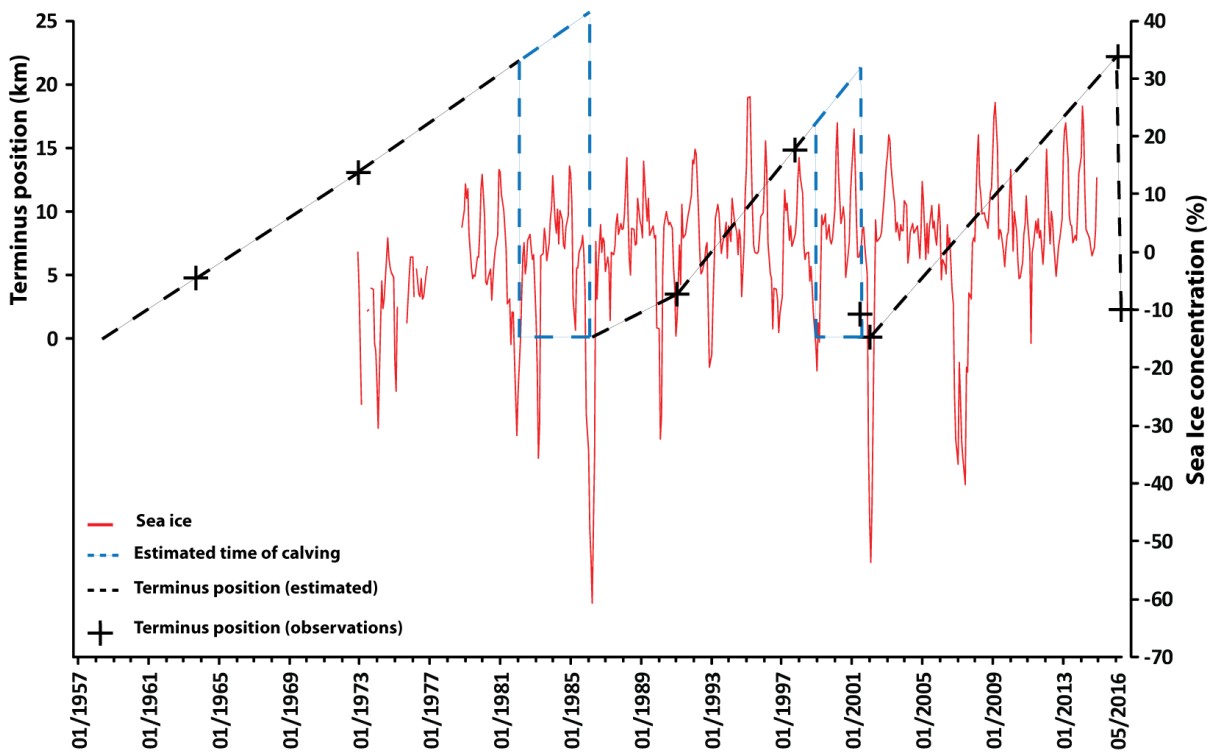

**Figure 12:** Reconstruction of the calving cycle of Holmes (West) Glacier. All observations are represented by black crosses. The estimated terminus positon is then extrapolated linearly between each observation. In periods without observations the date of calving is estimated by negative sea-ice concentration anomalies.

671

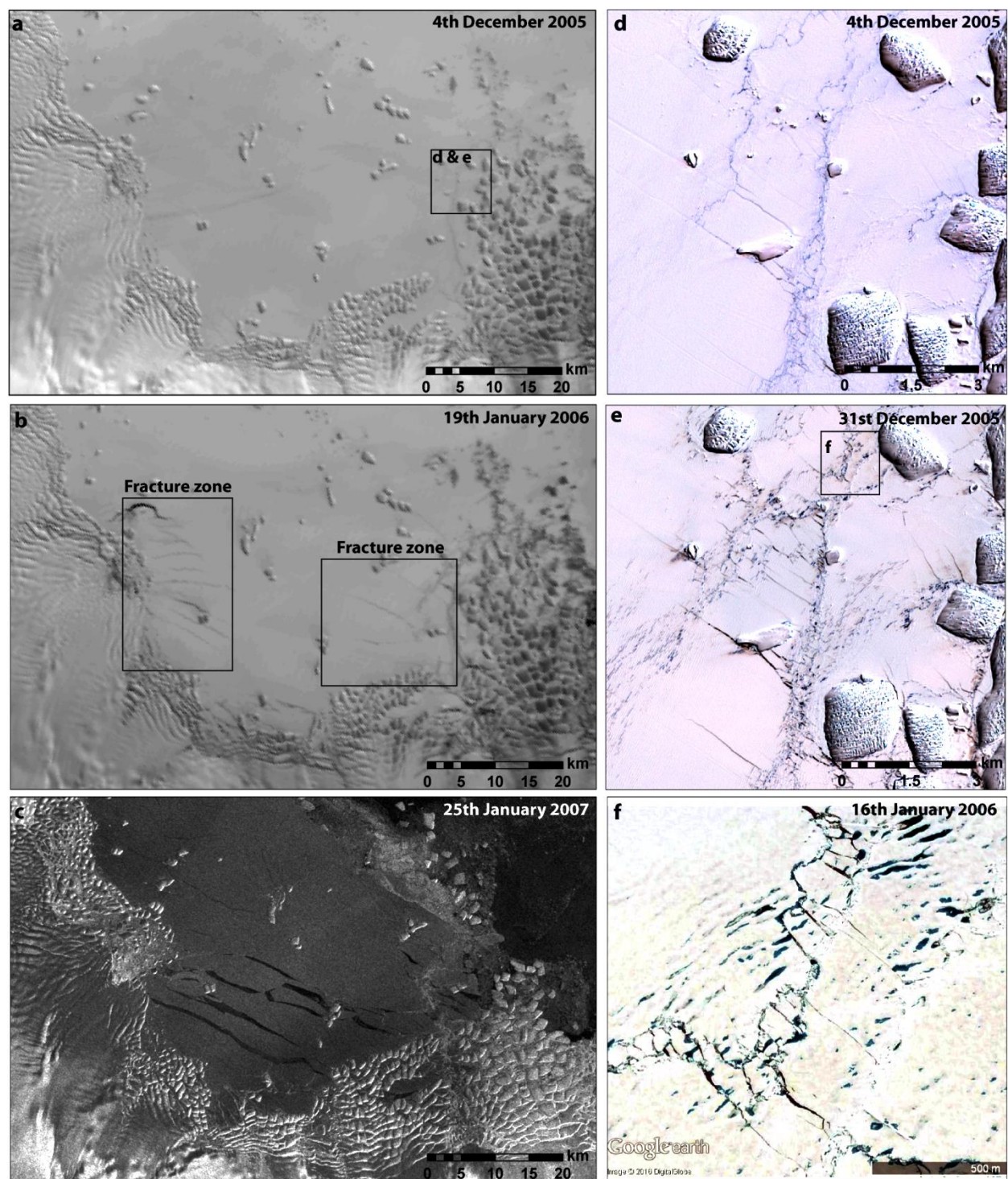

672

Figure 13: a and b) MODIS imagery showing the development of fractures in the landfast sea-ice between 4$^{th}$ December 2005 and 19$^{th}$ January 2006 (http://dx.doi.org/10.7265/N5NC5Z4N.) c) The landfast sea-ice ruptures along some of the same fractures which formed in December/January 2005/06, eventually leading to complete break-up in January 2007. d and e) ASTER imagery showing surface melt features and the development of smaller fracture between 4$^{th}$ and 31$^{st}$ December 2005. f) High resolution

optical satellite imagery from 16[th] January 2006 showing sea-ice fracturing and surface melt
ponding. This image was obtained from Google Earth.



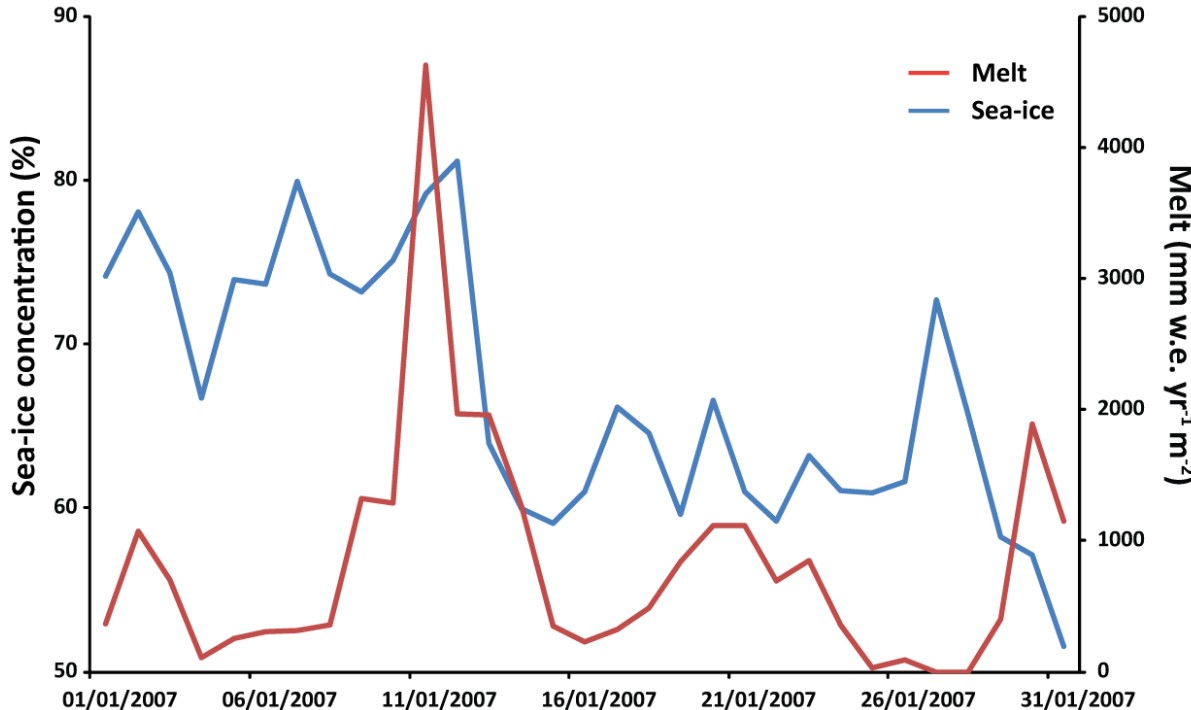

**Figure 14:** Daily sea-ice concentrations and RACMO derived melt during January 2007 in
Porpoise Bay. Sea-ice concentrations start to decrease after the melt peak on January 11[th].






