# Peer review of "Title: Simultaneous disintegration of outlet glaciers in Porpoise Bay (Wilkes Land),"

_The Cryosphere, 2016_

## Referee Comment (RC1) · A. Pope (Referee) · 23 Jul 2016

In "Simultaneous disintegration of outlet glaciers in Porpoise Bay (Wilkes Land), East Antarctica, and the long-term speed-up of Holmes Glacier," Miles et al. present a study which investigates a few East Antarctic tidewater outlets. They document calving activity with a range of remotely sensed imagery and relate calving behavior to sea ice dynamics. Using sea ice concentration as a proxy, they infer recent glacier speedup of 50%. This is an interesting paper, expanding a known glacial (in)stability mechanism from the Arctic to East Antarctica. With some revisions, I certainly recommend it for

publication in The Cryosphere. -Allen Pope

Broad Comments: **The paper leads to some really interesting points, but some of these are let down by incomplete discussions. The paper would be improved and more useful if the discussion of (1) what could have led to increased glacier velocity and (2) what drove anomalously high melt / ponding were expanded.

**Similarly, the conclusions are acceptable but do not place the results in a broader context, including the implications of the new knowledge described in the paper.

**This paper includes a lot of figures which help demonstrate and illustrate the arguments in the paper. This is really helpful! However, the figures are often complex imagery – more annotation would help the reader quickly understand what they are supposed to glean from a particular figure. In addition, many figures' brightness and contrast need to be reviewed for "readability" on screen and in print.

Specific Comments: L13: Indicate specific kinds of remote sensing data that were used.

L25: Include a space between "March" and "2016"

Abstract: The discussion includes mentions of warming, increased melting, etc. – including a sentence which nods to climate and larger implications may strengthen the abstract.

L93: The description of the method is VERY vague. What sort of automated mapping method? The goal should be reproducible science, so a fully described method should be included in the paper. At the very least, a citation which describes the method in depth should be included.

L126: 18 grid cells equals what area?

L131: Define ASI acronym

~L133: You discuss multiple breakouts – why is only 2007 studied at higher resolutions,
and are you sure the data sources are completely intercomparable?

L138: I thought that the figure described a particular region where sea ice concentration was studied. When/why are you getting closer to termini?

L141: At 27 km spatial resolution, how many points are you really sampling?

L155-L157: A bit of a meandering sentence, it almost implies monotonic behavior, which is not the case.

L159 & L167: It seems like Frost might not actually fit? More like a hybrid with Sandfjord?

L200: Using anomalies rather than absolute concentrations or areas means that sea ice could be lower, but it doesn't actually provide proof that there is a breakup.

L208: Are you really confident enough to use "cannot", as opposed to the slightly more flexible "likely did not"?

L211: Instead of "large," how about "very large" or "largest"?

Section 4.4: The first few paragraphs in this section seems more awkward and convoluted than previous sections. The sentence structure and tense seems overly complicated. It would benefit from a style edit so that it flows easier and therefore is more easily comprehended.

L219: "has been" to "was"

~L251-264: The language in this paragraph seems a bit belabored and the arguments (regarding sea ice) seem a bit circular. Streamline the writing to simplify and clarify. (On a side note: "thus" is repeated closely together, which is also awkward.)

L291: Consider including inferred velocities for these time periods, too?

L302: Insert space before open parenthesis

L308: Okay, it may be the first time it is observed. So what?

L313: "suggest", not "suggests"

L316: Days/weeks is really the realm of weather not climate – clarify the difference between the two and really what the important processes are.

~L322: Temperature might not be driving melt, but something in the model clearly is driving melt. Look at other parameters to identify this. For example, is it wind that could be causing it? That would be logical, and really helpful to identify the driver of such an important process.

L332: It may be the first time this has been published explicitly – but it is also not surprising. There are a few papers that observe supraglacial lakes on East Antarctic outlets. So why is it important that this has been observed for the first time?

~L347: Is it possible that the higher melt year saturated/refroze in the snowpack, which then allowed a lower melt year to be able to form melt ponds? I know that is the case on ice shelves, but I'm not sure if that is true in a sea ice context?

~L362: The sentences around here go in a couple circles about the processes and drivers that you think are most important for the reader to understand. I think it might help to clarify that, in this system, bathymetry and geometry seem to drive the location of calving events which sea ice drives the timing.

L376: This is restating earlier conclusions. Maybe only need to say in one place?

L380-389: This is really interesting and important glaciologically! The paper would be stronger if this were fleshed out and done so with more rigor. It can very much be a discussion of what is reasonable (not an in-depth analysis), but more should be included. For example, what might changed in accumulation do? Is it possible basal changes played a role? What else could be driving increased velocity?

~L399: Yes, sea ice is related to climate – but Antarctic sea ice is very much dependent on more than temperature (which can be seen in regional expansion of Antarctic sea ice). More nuance needs to be brought to this sentence.

L409: You specifically mention "warming" – but it would seem to be more appropriate to discussions in atmospheric or oceanic circulation?

L411: Okay – but where else might these processes be important? Expand this conclusion to be broader to have larger implications.

Table 2: No Landsat 8 OLI imagery used? This might be interesting for the recent breakup and data are available from 2013.

Figure 1: **Include a small inset of the entire continent. ***"Moscow University" should be "Moscow University Ice Shelf" **Scale bar in upper figure

Figure 2: **x-axis labels are a little too small **Caption should note the different vertical scales

Figure 3: **Blue is a bit hard to see **Show outline of this area in Figure 1? Don't worry about it if too crowded. **Brighten figure so easier to view

Figure 4: **Consider tracing front in a 2nd color in each image to clarify the changes that you want to highlight between images? It is hard to see (as you admit) with the melt, etc. **You reference the total area calved. Maybe include a hatched area in the last image between the two terminus lines?

Figure 6: Increase brightness and contrast to make more easily viewable.

Figure 7: Anomalies are interesting but is an absolute scale better to demonstrate what you want show?

Figure 8: Include 2nd outline in lower image?

Figure 9: **Maybe darken a little so it prints better? **Include 2nd outline in lower image?

Figure 10: Increase contrast so more viewable. The edge of the 9 km advance isn't very visible when printed.

Figure 11: Increase contrast in lower image.

Figure 12: Same comment as in text – include inferred velocities for these time periods, too?

Figure 13: Add line for 7 Feb? Hatched area to indicate calved area?

Figure 14: **Include annotation in each image and particularly at circle to help the reader **Increase contrast to make more viewable.

Figure 16: Maybe just report January '14 melt total relative to 2007? I don't think that the timeseries is particularly helpful here.

Figure 17 & 18: Combine these into one figure?

Figure 18: **Be consistent with date format **Double check permissions and copyright for using a Google Earth image in this publication.

---

## Referee Comment (RC2) · T.ÂăA. Scambos (Referee) · 26 Jul 2016

This is an interesting but rather rambling study on the connection between the presence of fast ice and the timing of glacier calving, and its potential impact on glacier flow. The study uses MODIS image data and the record of sea ice concentration from SSM/I (mostly) to demonstrate that during the brief periods of sea-ice-free conditions in Porpoise Bay, significant calving occurs, implicating the fast ice as a stabilizing component. The study then continues to search backward in time for evidence of this relationship, to the earliest satellite data, and then forward in time to an event in early 2016.

[Figure]

It could be published as it is... it is not incoherent. But it does not offer a clear contribution beyond the initial worthwhile documentation that fast-ice break-up leads to rapid iceberg release; or, to say it conversely, the presence of fast ice inhibits calving and drift. The discussion of velocity change seems rather vague, since any speed-up is not able to be directly tied to fast ice break-out and increased calving. The link to surface melting is so tenuous as to be useless. Moreover, the writing style is not brisk and efficient. There is a lot of excess text.

The central discovery of the study appears to be a major glacier calving event in late summer of 2007, during a period of extended sea ice retreat in Porpoise Bay. Images before, during and after the break-out of the sea ice show the disaggregation of the floating ice tongues in the region. Having found this, the authors extent the search, first backward in time through the sea ice record, and then back still further using the record of early Landsat and declassified Argon / Corona data.

There is another data set available, already processed, of MODIS data. See http://nsidc.org/data/iceshelves_images/index_modis.html This is a processed geolocated record of ~monthly to weekly images going back to 2000. This shows other periods of low sea ice and partial fast ice break-up. Unfortunately, there is a one-year gap around 2001-2002 that could help narrow down the 2002 calving. If you request some additional images, we can add them for this time range.

This manuscript is very long, and could be much better focused. It is not necessary to show every data set that can say something about the 2007 event – the purpose is to document the event and the link with fast ice break-out, and then examine the extent to which glacier flow might change as a result. In particular, it is lengthy to read, first, the discovery of the 2007 event, then the inference of the 2002 and and 1986 events, and then the possibility that there was no calving between 1963 and 1973 (and that case is not well-made) and then that there was a recent event.

The paper should present what you've learned overall, not the step-by-step process by
which you learned it.

I don't want to spend more time with the study. Nor do I want to berate the writers. So let me simply outline the paper they should write, if they wish to. Figure order is re-arranged. Any figure not listed is not needed.

Introduce the region (Figure 1b) Indicate the location of later figures here. Present the evidence for major break-ups in March 2016 and February 2007: (fig 13);(merge Figure 3-dates?- into Fig 4);(then, Fig2 extended through 2016). Fig6 might be retained, but it does not really help - regional sea ice is not low, yet there is a calving in austral autumn 2006.

Present the sea ice record from SSM/I+SMMR+EMSR for the areas in Fig 1b (using Figure 7) The major bay-wide calving and retreat is clearly timed by the loss of sea ice.

Earlier events large events at the Holmes Glacier front can also be linked, somewhat tentatively to extreme sea ice lows. MODIS data at NSIDC could help greatly here. An examination of this image series could document that the small iceberg pattern does / does not remain fixed (spreading or twisting of the arrangement of the bergs, but no 'individual' motion) during periods of continuous fast ice. Create a new figure from the MODIS data, summarizing what it shows about the small berg pattern evolution over time (this has a bit of Fig 14, but might be more extensive). Don't have to show every event in the 2000-2016 record – just the facts you extract from it.

With this record, i.e. ASAR, sea ice concentration, and MODIS, you have observations that can help you interpret the old record. Create a figure of the 1997, Nov 1973, Jan 1973, and Oct 1963 images, together. Given the field of small bergs trapped in fast ice in this series, make inferences about past calvings of Holmes, and of Frost (Figure 12). I would downplay the calvings of the other small glaciers, it is too confusing to follow it all.

From here, discuss synoptic patterns using reanalysis data that cause (or are associated with) sea ice retreat from this region of Antarctica. I don't think surface melting helps much, and in any case RACMO is not likely to capture katabatic heating events very well, which could be key in the immediate vicinity of the grounding line. I don't see how Fig 15, 16, 17 or 18 really helps.

This will be a new paper, that you would re-submit for review. It would be half as long. It would conclude things about calving and presence of fast ice, mostly focused on the the 2016, 2007, and 2002 events, but with a few statements about earlier events back to 1963. It would conclude things about berg motion within temporally continuous fast ice, and synoptic climate patterns that favor fast ice break-out and calving in this area.

Please also note the supplement to this comment:
http://www.the-cryosphere-discuss.net/tc-2016-151/tc-2016-151-RC2-supplement.pdf

———————————————————

[Figure]

**Supplement:**

[Figure]

[Figure]

**Title: Simultaneous disintegration of outlet glaciers in Porpoise Bay (Wilkes Land),**

**East Antarctica, and the long-term speed-up of Holmes Glacier.**

**Authors:** B.W.J. Miles[*], C. R. Stokes, S.S.R. Jamieson

**Affiliation:** *Department of Geography, Durham University, Science Site, South Road, Durham, DH1 3LE,*

*UK*

*\*Correspondence to: a.w.j.miles@durham.ac.uk*

**Abstract: The floating ice shelves and glacier tongues which fringe the Antarctic**

**continent are important because they help buttress ice flow from the ice sheet interior.**

**Dynamic feedbacks associated with glacier calving have the potential to reduce**

**buttressing and subsequently increase ice flow into the ocean. However, there are few**

**high temporal resolution studies on glacier calving, especially in East Antarctica. Here**

**we use remote sensing to investigate monthly glacier terminus change across six marine-**

**terminating outlet glaciers in Porpoise Bay (-76°S, 128°E), Wilkes Land (East**

**Antarctica), between November 2002 and March 2012. This reveals a large**

**simultaneous calving event in January 2007, resulting in a total of ~2,900 km² of ice**

**being removed from glacier tongues. Our observations suggest that sea-ice must be**

**removed from glacier termini for any form of calving to take place, and we link this**

**major calving event to a rapid break-up of the multi-year sea-ice which usually occupies**

**Porpoise Bay. Using sea-ice concentrations as a proxy for glacier calving, and by**

**analysing available satellite imagery stretching back to 1963, we reconstruct the long-**

**term calving activity of the largest glacier in Porpoise Bay: Holmes (West) Glacier. This**

**reveals that its present-day velocity (~1450 m a⁻¹) is approximately 50% faster than**

**between 1963 and 1973 (~900 m a⁻¹). We also observed the start of a large calving event**

**in Porpoise Bay in March 2016 that is consistent with our reconstructions of the**

**periodicity of major calving events. These results highlight the importance of sea-ice in**

**modulating outlet glacier calving and velocity in East Antarctica.**

[Figure]

**1. Introduction**

Iceberg calving is an important process that accounts for around 50% of total mass loss to the ocean in Antarctica (Depoorter et al., 2013; Rignot et al., 2013). Moreover, dynamic feedbacks associated with retreat and/or thinning of buttressing ice shelves or floating glacier tongues can result in an increased discharge of ice into the ocean (De Angelis and Skvarca, 2003; Rignot et al., 2004; Wuite et al., 2015). At present, calving dynamics are only partially understood (Benn et al., 2007; Chapuis and Tetzlaff, 2014) and models struggle to replicate observed calving rates (van der Veen, 2002; Astrom et al., 2014). Therefore, improving our understanding of the mechanisms driving glacier calving and how glacier calving cycles have responded to recent changes in the ocean-climate system is important in the context of future ice sheet mass balance and sea level.

Calving is a two-stage process that requires both the initial ice fracture and the subsequent transport of the detached iceberg away from the calving front (Benn and Jacobs, 2013). In Antarctica, major calving events can be broadly classified into two categories: the discrete detachment of large tabular icebergs (e.g. Mertz glacier tongue: Massom et al., 2015) or the spatially extensive disintegration of floating glacier tongues or ice shelves into numerous smaller icebergs (e.g. Larsen A & B ice shelves (Rott et al., 1996; Scambos et al., 2009). Observations of decadal-scale changes in glacier terminus position in both the Antarctic Peninsula and East Antarctica have suggested that despite some degree of stochasticity, iceberg calving and glacier advance/retreat is likely driven by external climatic forcing (Cook et al., 2005; Miles et al., 2013). However, despite some well-documented ice shelf collapses (Scambos et al., 2003; Banwell et al., 2013) and major individual calving events (Masson et al., 2015) there is a paucity of data on the nature and timing of calving from glaciers in Antarctica (e.g. compared to Greenland: Moon and Joughin, 2008; Carr et al., 2013), and particularly in East Antarctica.

Following recent work that highlighted the potential vulnerability of the East Antarctic Ice Sheet in Wilkes Land to ocean-climate forcing and marine ice sheet instability (Greenbaum et al., 2015; Aitken et al., 2016; Miles et al., 2016), we analyse the recent calving activity of six outlet glaciers in the Porpoise Bay region using monthly satellite imagery between November 2002 and March 2012. We then turn our attention to investigating the drivers behind the observed calving dynamics, before examining evidence for any longer term changes in

calving using sea ice concentrations and satellite imagery from 1963, 1973, 1991, 1997, 2002

and 2016.

[revised manuscript text omitted]
 calving event described in the previous section (see Fig. 4), and strongly suggesting that the two processes are linked. The series of satellite images showing the evolution of the January to April 2007 calving event clearly shows glacier calving taking place after initial sea-ice breakup e.g. Fig. 4b-e. Furthermore, the smaller calving events of

Sandford and Frost glaciers all take place after sea-ice had retreated away from the glacier terminus (Fig. 6). Indeed, throughout the study period, there is no evidence of any calving events taking place with sea-ice proximal to glacier termini. This suggests that glaciers in

Porpoise Bay are very unlikely to calve with sea-ice present at their termini.

**4.4 Longer-term glacier calving cycles**

We now turn our attention to reconstructing calving activity and glacier frontal position change over a longer time-period, with a particular focus on the largest glacier – Holmes (West). Our terminus position change results indicate that glaciers in Porpoise Bay will only calve when sea-ice breaks away from glacier termini. Analysis of long-term sea-ice concentrations in

Porpoise Bay from 1972 to 2014 suggests that there have been larger sea-ice break-up events prior to January 2007 (Fig. 7). The two largest break-up events occurred in April 1986 and

February 2002, when monthly sea-ice concentrations suggest a near-complete removal of all sea-ice in the Bay, unlike in January 2007, where sea-ice remained in the west section of the bay in front of Holmes (West) Glacier (Fig 4). This suggests that the only time Holmes (West)

Glacier's terminus was free of sea-ice during our observational period (from 1972-2014) was in April 1986 and February 2002. Moreover, although there are several other moderate negative monthly mean sea-ice anomalies (~20 to 30%) throughout the sea-ice concentration observational period (Fig. 7), we suggest these cannot have resulted in the Holmes West

Glacier terminus being sea-ice free. For its terminus to be clear of sea-ice, the sea-ice in the outer regions of Porpoise closest to the open ocean must be removed before the sea-ice close to its terminus. Therefore, it is only the large sea-ice anomalies which can result in the

Holmes (West) Glacier terminus being sea-ice free i.e. the removal of all sea-ice in the bay.

Thus, it is very likely Holmes (West) Glacier calved in April 1986 and February 2002. Ideally, we would test this by analysing a series of satellite images (e.g. Fig 4). However, because there is no cloud-free satellite imagery available around the time of its proposed calving periods (April 1986 and February 2002), we rely on a comparison between satellite images that are as close as possible to before and after the major sea-ice break-up events.

[Figure]

By analysing available satellite imagery from October 1997 and August 2002 (see Fig. 8), it is clear that there has been a large calving event at Holmes (West) Glacier at some point between these dates. This is because the August 2002 position is around 15 km behind the October 1997

position (Fig. 8b). As noted above, our observations of sea-ice concentrations (Fig. 7) suggest that the most likely time would be in February 2002, which is the only major negative sea-ice anomaly that might have been large enough to indicate an absence of sea-ice in front of the glacier's terminus. This is further supported by observations of Holmes (West) Glacier calving front in August 2002 (i.e. little crevassing) (Fig. 8b), which is entirely consistent with a calving event having taken place a few months beforehand.

The nearest available satellite imagery either side of the April 1986 sea-ice break-up event is in

January 1973 and February 1991 (Fig. 9) and, again, it is clear from the position of the glacier terminus in February 1991 that there has been a calving activity at some point between these dates, which we suggest occurred in April 1986 based on the major negative sea-ice concentration data. Indeed, the terminus position of Holmes (West) Glacier in February 1991 is entirely consistent with a calving event in April 1986, assuming that it calves to a similar position following each calving event e.g. perhaps losing the unconstrained section of its glacier tongue. That is, if the glacier calved in April 1986, as we suggest, we would expect it to have advanced by the time of the next available image in February 1991 (Fig. 9b). Therefore, we suggest that these observations are entirely consistent with two major calving events at

Holmes (West) Glacier in April 1986 and February 2002. We now turn our attention to extending this record by analysing imagery from 1963 and 1973.

The 1963 ARGON satellite image shows Holmes (West) Glacier terminus (and indeed most of the glacier termini) very close to the August 2002 position, which we suggest is just a few months after a major calving event in February 2002. Thus, the 1963 image might suggest that

Holmes (West) Glacier (and other glaciers) had recently calved prior to 1963 (Fig. 10a). By

January 1973, however, Holmes (West) Glacier had advanced around 9 km from its 1963

position (Fig. 10b). Given Holmes (West) Glacier's present-day velocity of ~1,400 m yr$^{-1}$

(Rignot et al., 2011b), an advance of around 14 km would be expected in the ten year period between 1963 and 1973. This means that Holmes (West) Glacier either advanced at a slower rate in the 1960s (~900 m yr$^{-1}$) or that the glacier calved between 1963 and 1973. Analysis of the 1963 and 1973 images suggests that calving activity is unlikely. This is because individual icebergs can be tracked from the front of Holmes (West) Glacier in 1963 to the edge of the multi-year sea ice pack in 1973 (Fig. This confirms that there has been no sea-ice break-up

events and, as such, no major calving events between 1963 and 73. Furthermore, Sandford

Glacier tongue can be seen to advance several kilometres between October 1963 (Fig. 10a) and

November 1973 (Fig. 10b). If there had been a sea-ice break-up, this ice tongue would likely calved and been transported away from the terminus. Moreover, in all available satellite imagery after 1973, the largest glacier tongue observed at Sandford glacier is only ~2 km. As

Sandford Glacier is the closest glacier to the open ocean in Porpoise Bay, its terminus can be sea-ice free even during relatively small sea-ice break-up events. Therefore, in order to facilitate the growth of a ~10 km glacier tongue between 1963 and 1973 (Fig. 10), it suggests that there must have been high sea-ice concentrations in Porpoise Bay during this period, thus helping to preserve Sandford Glacier tongue. Thus, we suggest that it is highly unlikely that any glaciers calved in Porpoise Bay between 1963 and 1973 because there were no sea-ice break-up events. This implies that the velocity of the Holmes West Glacier between 1963 and

1973 was slower (~900 m yr$^{-1}$) during that era, and that the glacier velocity has approximately increased by 50% since that time.

Combining the known terminus position with the velocity estimates between 1963 and 1973, and the calving events in April 1986 and February 2002, allows us to reconstruct the long-term calving cycle of Holmes (West) Glacier (Fig. 12). In order to do this we make two assumptions. First, we simply extrapolate velocity linearly in between periods without observations. Secondly, to determine how far the terminus retreated after the calving event in

1986 and the date of calving before 1963, for which we have no imagery, we simply assume it retreats close to the position attained after the February 2002 calving event in August 2002.

Our reconstruction suggests that, despite an increase in velocity, Holmes (West) Glacier tends to calve when its terminus reaches an extended position that is around 20 km from its known retreat positions in 1986 and 2002. Furthermore, we note that the very recent terminus position (austral summer 2016) is in a similar position to that which existed immediately prior to the calving events of April 1986 and February 2002, suggesting that a further major calving event is imminent.

**4.5 2016 calving event**

During the preparation of this manuscript, and consistent with our conclusion from the previous section, observations between March 19th and May 13th 2016, revealed that Frost glacier, Holmes (East) and Holmes (West) glaciers underwent a further disintegration event following the break-up of sea-ice from their glacier tongues (Fig. 13). This process has so far

resulted in the loss of ~1,500 km$^2$ of ice from glacier tongues in Porpoise Bay. The calving event is likely incomplete and may continue, potentially also influencing Glacier 1 and 2. We note that the recent calving of Holmes (West) Glacier is entirely consistent with our earlier observations in that: 1) sea-ice must be removed in order for Holmes (West) Glacier and other glaciers in Porpoise Bay to calve (Fig.14); 2) Holmes (West) glacier undergoes a major calving event after reaching a similar position in each calving cycle (e.g. Fig. 12); 3) Holmes (West)

glacier retreats to a similar position after each calving event. Furthermore, we can now estimate that the previous three calving cycles of Holmes West glacier have been in ~29

(~1957-1986), 16 (1986-2002) and 14 (2002-2016) year cycles.

**5. Discussion**

**5.1 Climatic drivers of the January 2007 calving event**

We report a major, synchronous calving event in January 2007 that resulted in ~2,900 km$^2$ of ice being removed from glacier tongues in the Porpoise Bay region of East Antarctica. This is comparable to some of the largest disintegration events ever observed in Antarctica e.g. Larsen

A, 1995 (4,200 km$^2$), Larsen B, 2002 (3,250 km$^2$), and is the largest to have been observed in

East Antarctica. However, this event differs to those observed on the ice shelves of the

Antarctic Peninsula, in the sense that it is more closely linked to a predictable cycle of glacier advance and retreat (e.g. Fig. 12), as opposed to a catastrophic collapse that may be unprecedented. That said, it is intriguing that there is evidence of this cycle speeding up over the past 50 years, concomitant with an increase in glacier velocity(e.g. Fig. 12)

The disintegration event was driven by the break-up of the multi-year land-fast sea-ice which usually occupies Porpoise Bay. This link between sea-ice and glacier terminus position has been largely confined to studies in Greenland, where sea-ice melange dynamics has been linked to inter-annual variations in glacier terminus position (Amundson et al., 2010; Carr et al., 2013; Todd and Christoffersen, 2014; Cassotto et al., 2015). However, this is the first time sea-ice has been linked to large scale disintegration of glacier tongues in East Antarctica.

It is likely that multiple climatic processes operating over different timescales contributed to the January 2007 sea-ice break-up event. This is because the majority of sea-ice in Porpoise

Bay is multi-year sea-ice (Fraser et al., 2012). Although there are no long-term observations of multi-year sea-ice thickness in Porpoise Bay, observations and models of the annual cycle of multi-year sea-ice in other regions of East Antarctica suggests that multi-year sea-ice thickens seasonally and thins each year (Lei et al., 2010; Sugimoto et al., 2016; Yang et al., 2016).

[Figure]

Therefore, the relative strength, stability and thickness of multi-year sea ice at a given time period is driven not only by climatic conditions in the short term (days/weeks), but also by climatic conditions in the preceding years.

As the sea-ice break-up occurred during the peak of austral summer in January 2007, it is plausible that air temperature played an important role in initiating the sea-ice break-up.

Analysis of RACMO2.3 mean monthly melt values in Porpoise Bay show that although

January 2007 was above the average, it was not exceptional, lying within one standard deviation of the long term mean (1979-2015). However, analysing daily melt values throughout January 2007 suggests that there was an exceptional melt event centred on the 11$^{th}$

January (Fig. 15). This melt is the 11$^{th}$ highest day on record (1979-2015) and the 4$^{th}$ highest since 2000. Analysis of daily sea-ice concentrations in Porpoise Bay show an immediate drop after this melt peak (Fig. 15), suggesting the exceptional melt peak of the 11$^{th}$ January may have been important in initiating sea-ice break-up. As a consequence of a melt peak of this magnitude, the growth of sea-ice surface ponding would be expected. There is no cloud-free optical satellite imagery available for January 2007 to confirm this prediction. However,

Landsat imagery from the 21$^{st}$ January 2014, which occurs shortly after a melt event of a similar magnitude, clearly demonstrates that substantial sea-ice melt ponding is possible near the coast in Porpoise Bay (Fig. 16). Indeed, this is the first time that sea-ice ponding to this extent has been observed in coastal East Antarctica. In the Arctic, sea-ice melt ponding along pre-existing weaknesses has been widely reported to precede sea-ice break-up (Ehn et al.,

2011; Petrich et al., 2012; Landy et al., 2014; Schroder et al., 2014; Arntsen et al., 2015).

However, because there have been similar magnitude melt events to that of mid-January 2007

which have not resulted in the break-up of sea-ice in Porpoise Bay, we suggest that whilst it may have driven the initial sea-ice break-up, it was probably dependent on other preceding factors.

In the austral summer melt season (2005/06) that preceded the break-up event in January 2007, there was an anomalously high mean melt in December 2005 (Fig. 17). Indeed, December

2005 ranks as the second warmest month on record (1979-2015) in Porpoise Bay. To place this month into perspective, we note that it would rank above the average melt value of all

Decembers and Januarys since 2000 on the remnants of Larsen B ice shelf. High resolution optical satellite imagery reveal extensive sea-ice melt ponding and fracturing following this melt event in January 2006 (Fig. 18), and it is plausible that this exceptionally warm month may have weakened the multi-year sea-ice in Porpoise Bay and primed it for break-up the

following year. Indeed, by the end of the 2005/06 melt season, the sea-ice pack in Porpoise

Bay had retreated to the edge of Frost Glacier (e.g. Fig 6), suggesting that the sea-ice may have come close to complete break-up. Therefore, we hypothesise that the January 2007 sea-ice break-up event was driven by a combination of an exceptionally warm 2005/06 austral summer, which caused weakening of multi-year sea-ice, but with break-up initiated the following melt season after the January 11[th] melt event.

**5.2 Calving cycle and increase in velocity of Holmes West Glacier**

Our reconstruction of the calving cycle of Holmes (West) Glacier (Fig. 12) indicates that the glacier undergoes a major calving event when it reaches roughly the same position in each cycle. This suggests that calving is likely to be influenced by the bathymetry and topography of Porpoise Bay. However, sea-ice must still be removed in order for Holmes (West) Glacier to calve, suggesting a complex interaction between the stability of Holmes (West) Glacier's floating tongue, bathymetry, topography and sea-ice. In both Greenland (McFadden et al.,

2011; Carr et al., 2013; Carr et al., 2015) and Antarctica (Wang et al., 2016), underlying bathymetry is thought to be crucial in determining the calving of floating glacier tongues.

However, our results suggest that the bathymetry and topography of Porpoise Bay may only be a secondary control to the calving of Holmes (West) Glacier. This is because sea-ice must be removed from its terminus before calving. Indeed, we note that complete removal of sea-ice from Porpoise Bay only occurs when Holmes (West) Glacier is at an advanced position. If the break-up of sea-ice was solely driven by climate, complete break-ups would be expected under strong climatic warming events, irrespective of the position of Holmes (West) Glacier.

Therefore, we speculate sea-ice break-ups must be at least in part influenced by the position of

Holmes (West) Glacier tongue itself. That is, as Holmes (West) Glacier advances it slowly pushes multi-year sea-ice further out into the open ocean to the point where the multi-year sea- ice pack may become unstable. This could be influenced by local bathymetry and ocean circulation, but no observations are available. However, we note that once the glacier forces the sea-ice into a more unstable region, it still requires a strong climatic warming event to initiate the sea-ice break-up (see section 5.1) and subsequent glacier calving.

Despite Holmes (West) Glacier consistently calving in approximately the same position, the time taken for the glacier to calve in each cycle has decreased, demonstrating an increase in glacier velocity. Indeed, our estimates suggest that the present day-velocity of Holmes (West)

Glacier is approximately 50% faster than its average 1963-1973 velocity. This is significant

because, based on the flux gate calculations of Rignot et al. (2013), Holmes (West) Glacier is now exporting approximately 8 GT yr$^{-1}$ more into the ocean than it was between 1963 and

1973. This also provides the first evidence of a long term increase in velocity of an outlet glacier in East Antarctica. A potential mechanism which could explain this increase in velocity is changes to the stability and strength of the sea-ice in Porpoise Bay reducing glacier buttressing. Alternatively, dynamic changes associated with incursions of warm subsurface ocean water and associated thinning could have driven the increase in velocity e.g. Pine Island

Glacier (Rignot, 2008; Jacobs et al., 2011). However, with sea-ice concertation data only available after 1972, and with only limited atmospheric data, and no oceanic or sea-ice thickness data, it is impossible to be more conclusive.

**6. Conclusion**

Glacier terminus position changes are analysed at approximately monthly intervals between

November 2002 and March 2012 for six glaciers in Porpoise Bay, Wilkes Land, East

Antarctica. We identify a large simultaneous calving event in January 2007 which was driven by the break-up of the multi-year landfast sea-ice which usually occupies the bay. This provides a previously unreported mechanism for the rapid disintegration of floating glacier tongues in East Antarctica. Throughout the observational period, major calving activity only takes place following the near-complete removal of sea-ice from glacier termini. This is an important discovery because sea-ice and land-fast sea-ice are widely considered to be highly sensitive to changes in climate (Heil, 2006; Mahoney et al., 2007). Therefore, if the sea-ice which usually occupies Porpoise Bay became weaker or less permanent in a warmer climate, there could be an associated dynamic response of glaciers following the decrease in buttressing.

Reconstructions of the calving cycle of Holmes (West) Glacier show that its present day velocities are approximately 50% faster than between 1963 and 1973, making it the only glacier in East Antarctica known to exhibit a recent increase in velocity. As the interaction between sea-ice and floating glacier tongues is currently poorly represented in models, we suggest that this may provide another mechanism capable of explaining some of the rapid mass loss which may have happened in the past, and may be an important process in the context of future warming. We conclude by highlighting the importance of regular monitoring of glaciers in Porpoise Bay following the 2016 calving event, and in particular, the re-formation of the landfast ice following its break-up.

[Figure]

**Acknowledgements:** We thank the ESA for providing Envisat ASAR WSM data (Project ID:
16713) and Sentinel data. Landsat imagery was provided free of charge by the U.S. Geological
Survey Earth Resources Observation Science Centre. We thank M. van den Broeke for
providing data and assisting with RACMO2.3. B.W.J.M was funded by a Durham University
Doctoral Scholarship program. S.S.R.J. was supported by Natural Environment Research
Council Fellowship NE/J018333/1.

[Figure]

**Table 1:** Glacier velocities from Rignot et al. (2011b)

| Glacier | Velocity (m yr$^{-1}$) |
|---|---:|
| Sandford | 440 |
| Frost | 2000 |
| Glacier 1 | 950 |
| Glacier 2 | 500 |
| Holmes (East) | 600 |
| Holmes (West) | 1450 |

**Table 2: Satellite imagery used in the study**

| Satellite | Date of Imagery | Spatial resolution (m) |
|---|---|---|
| ARGON | October 1963 (Kim et al., 2007) | 140 |
| Envisat ASAR WSM | August 2002, November 2002 to March 2012 (monthly) | 80 |
| Landsat (MSS) | January 1973 | 60 |
| Landsat (TM) | February 1991 | 30 |
| MODIS | March 2016 | 250 |
| RADARSAT | September 1997 (Liu and Jezek, 2004) | 100 |
| Sentinel-1 | February-May, 2016 | 40 |

[Figure]

[Figure]

**Figure 1: a)** MODIS image of Wilkes Land, East Antarctica **b)** Landsat images of Porpoise Bay with glacier velocity (Rignot et al., 2011b) and grounding lines (Rignot et al., 2011a) overlain. The hatched polygon represents the region where sea-ice concentrations were extracted.

[Figure]

[Figure]

**Figure 2:** Terminus position change of six glaciers in porpoise Bay between November 2002
and March 2012. Note the major calving event in January 2007 for 5 of the glaciers.
Terminus position measurements are subject to +/- 500 m.

[Figure]

[Figure]

**Figure 3:** Envisat ASAR WSM imagery in January 2007 **a)** and April 2007 **b)**, which are immediately prior to and after a simultaneous calving event in Porpoise Bay. Red line shows terminus positions in January 2007 and blue line shows the positions in April 2007.

[Figure]

[Figure]

**Figure 4:** Envisat ASAR WSM imagery showing the evolution of the 2007 calving event.
Red line shows the terminus positions from December 11[th] 2006 on all panels.

[Figure]

[Figure]

[Figure]

**Figure 5:** Mean monthly sea-ice concentration anomalies in Porpoise Bay.

[Figure]

[Figure]

**Figure 6:** Time series of Frost and Sanford Glaciers calving showing that sea-ice clears prior to calving and dispersal of icebergs.

[Figure]

[Figure]

**Figure 7:** Mean monthly sea-ice concentration anomalies 1972-2014. Note major anomalies
in April 1986, February 2002 and January 2007.

[Figure]

[Figure]

[Figure]

**Figure 8:** Comparison of terminus positon between **a)** October 1997 (red line) and **b)** August 2002, which indicates major calving event(s) at some point between these two dates.

[Figure]

[Figure]

[Figure]

**Figure 9:** Comparison of terminus position change between January 1973 (blue line) and
February 1991, which indicates a calving event at some point between these two dates.

[Figure]

[Figure]

**Figure 10: a)** October 1963 terminus position. The red line shows the August 2002 terminus positon, which occurred a few months after a major calving event. Because Holmes (West) glacier (and other glaciers) is in a similar position, it suggests that there has been a calving event within a few years prior to this image i.e. late 1950s/early 1960s. **b)** November 1973 terminus positon in relation to 1963 (blue). The relative position of glacier in Porpoise Bay in 1973 suggests that there were no calving events between these dates.

[Figure]

[Figure]

[Figure]

[Figure]

**Figure 11:** Iceberg tracking in front of Holmes (West) Glacier. The same iceberg can be seen
in both 1963 and 1973 suggesting there has not been a sea-ice break-up event during this
period (see also Figure 10 and the floating tongue on Sandford Glacier)

[Figure]

[Figure]

**Figure 12:** Reconstruction of the calving cycle of Holmes (West) Glacier. All observations
are represented by black crosses. The estimated terminus positon is then extrapolated linearly
between each observation, with major calving inferred to coincide with major negative sea-
ice concentration anomalies in 1986, 2002 and 2016. This suggests the previous three calving
cycles to be ~29 years (~1957-1986), 16 years (1986-2002) and 14 years (2002-2016).

[Figure]

[Figure]

**Figure 13:** Time series of the (likely ongoing) evolution of the 2016 calving event in
Porpoise Bay using Sentinel-1 satellite imagery. The disintegration event starts at some point
between 2nd March and 26th March. By the 13th May Holmes (West) Glacier has retreated
approximately 20 km and ~1,500 km$^2$ of ice had been lost from glacier tongues in Porpoise
Bay.

[Figure]

[Figure]

[Figure]

[Figure]

**Figure 14:** MODIS imagery showing the initial stages of disintegration of Holmes (West)
Glacier in March 2016. On March 19[th] a large section of sea-ice breaks away from the
terminus, initiating the rapid disintegration process.

[Figure]

[Figure]

**Figure 15:** Daily sea-ice concentrations and RACMO2.3 derived melt during January 2007
in Porpoise Bay. Sea-ice concentrations start to decrease after the melt peak on January 11[th].

[Figure]

[Figure]

[Figure]

**Figure 16:** Evidence of substantial sea-ice surface ponding on the 21$^{st}$ January 2014 (arrows)
following the exceptional melt event centred on the 31$^{st}$ December.

[Figure]

[Figure]

[Figure]

**Figure 17:** Mean RACMO2.3 December melt 1979-2015 in Porpoise Bay.

[Figure]

[Figure]

**Figure 18: a)** Envisat ASAR WSM image from January 2006. **b, c, d)** High resolution
optical satellite imagery from 16/1/2006 showing sea-ice fracturing and surface melt ponds
following the exceptionally high melt in December 2005, which were obtained from Google
Earth.

---

## Author Comment (AC1) · 2 Oct 2016

**Response to Reviewers**

We would like to thank both referees for taking the time to provide detailed feedback on our manuscript. We are pleased that both referees found our discovery that large simultaneous calving events in Porpoise Bay are driven by the break-up of multi-year sea-ice interesting and worth publishing. The referees were less positive about section 4.4 (reconstructing calving cycles) and reviewer 2 suggested this should be removed and that the paper could be considerably shortened.

In line with reviewer comments, we have decided to remove nearly all of the discussion of longer-term calving cycles from the manuscript and re-structure the paper, which does not impact on our main findings. Additionally, we have added further discussion on the 2016 calving event which has progressed further since our initial manuscript submission.

The revised manuscript is shorter and contains fewer figures. In short, the first half of the paper has remained similar, the section on reconstructing glacier calving has been removed, and a more in depth analysis on the drivers of the January 2007 and March 2016 sea-ice break-up has been added, as requested by the reviewers.

We include specific replies in blue to each reviewer and also attach a revised manuscript with changes highlighted in blue:

**Reviewer 1**

In "Simultaneous disintegration of outlet glaciers in Porpoise Bay (Wilkes Land), East Antarctica, and the long-term speed-up of Holmes Glacier," Miles et al. present a study which investigates a few East Antarctic tidewater outlets. They document calving activity with a range of remotely sensed imagery and relate calving behavior to sea ice dynamics. Using sea ice concentration as a proxy, they infer recent glacier speedup of 50%. This is an interesting paper, expanding a known glacial (in)stability mechanism from the Arctic to East Antarctica. With some revisions, I certainly recommend it for C1 publication in The Cryosphere. -Allen Pope

We thank the reviewer for the positive comments regarding the manuscript.

Broad Comments: **The paper leads to some really interesting points, but some of these are let down by incomplete discussions. The paper would be improved and more useful if the discussion of (1) what could have led to increased glacier velocity and (2) what drove anomalously high melt / ponding were expanded. **Similarly, the conclusions are acceptable but do not place the results in a broader context, including the implications of the new knowledge described in the paper. **This paper includes a lot of figures which help demonstrate and illustrate the arguments in the paper. This is really helpful! However, the figures are often complex imagery – more annotation would help the reader quickly understand what they are supposed to glean from a particular figure. In addition, many figures' brightness and contrast need to be reviewed for "readability" on screen and in print.

Specific Comments: L13: Indicate specific kinds of remote sensing data that were used.

Amended: we have removed the section on long-term calving cycles and glacier velocity so the first point raised is no longer an issue. We address the second point by adding lengthy discussion of the possible causes of the anomalously high melt. We have checked all figures and annotated them for greater clarity.

L25: Include a space between "March" and "2016"

Amended.

Abstract: The discussion includes mentions of warming, increased melting, etc. – including a sentence which nods to climate and larger implications may strengthen the abstract.

The abstract has been changed to reflect the revised manuscript and mentions the wider implications re: increased melting, climate change.

L93: The description of the method is VERY vague. What sort of automated mapping method? The goal should be reproducible science, so a fully described method should be included in the paper. At the very least, a citation which describes the method in depth should be included.

We have updated the description of the method. The description now details how method automatically classifies glaciers and sea-ice into polygons based on the pixel statistics of each image.

L126: 18 grid cells equals what area?

Amended, 18 grid cells equates to 11,250 $km^{2}$.

L131: Define ASI acronym

Amended.

~L133: You discuss multiple breakouts – why is only 2007 studied at higher resolutions, and are you sure the data sources are completely intercomparable?

The manuscript no longer discusses multiple breakouts and the longer term calving cycle. Thus, we focus primarily on the 2007 break-out, but have also added some discussion of a very recent event that may have initiated in 2016.

L138: I thought that the figure described a particular region where sea ice concentration was studied. When/why are you getting closer to termini?

Amended. This was an oversight; we now also include the region where the higher resolution sea-ice data was extracted (which is closer to the terminus) in Figure 1 and mention this in the text.

L141: At 27 km spatial resolution, how many points are you really sampling?

Amended.

L155-L157: A bit of a meandering sentence, it almost implies monotonic behavior, which is not the case.

Amended.

L159 & L167: It seems like Frost might not actually fit? More like a hybrid with Sandfjord?

All calving events which we observe in Porpoise Bay only occur after sea-ice has broken away from glacier termini. We clarify this is in section 4.4.

L200: Using anomalies rather than absolute concentrations or areas means that sea ice could be lower, but it doesn't actually provide proof that there is a breakup.

We appreciate that anomalies alone do not provide proof of complete break-up. However, when combined with imagery actually showing the break-up, the events would appear to be validated. We could show absolute concentrations instead, but these are very difficult to follow, especially over multiple years due to the strong seasonal variations.

L208: Are you really confident enough to use "cannot", as opposed to the slightly more flexible "likely did not"?

Section removed in response to Reviewer 2 comments.

L211: Instead of "large," how about "very large" or "largest"?

Section removed in response to Reviewer 2 comments.

Section 4.4: The first few paragraphs in this section seems more awkward and convoluted than previous sections. The sentence structure and tense seems overly complicated. It would benefit from a style edit so that it flows easier and therefore is more easily comprehended.

This section has been removed in response to Reviewer 2 comments.

L219: "has been" to "was"

Section removed in response to Reviewer 2 comments.

~L251-264: The language in this paragraph seems a bit belabored and the arguments (regarding sea ice) seem a bit circular. Streamline the writing to simplify and clarify. (On a side note: "thus" is repeated closely together, which is also awkward.)

Section removed in response to Reviewer 2 comments.

L291: Consider including inferred velocities for these time periods, too?

Section removed in response to Reviewer 2 comments.

L302: Insert space before open parenthesis

Section removed in response to Reviewer 2 comments.

L308: Okay, it may be the first time it is observed. So what?

We now state its wider importance. If future changes in climate result in a weaker persistence in landfast ice in porpoise bay, it may result in detrimental effects on glacier tongue stability.

L313: "suggest", not "suggests"

Amended.

L316: Days/weeks is really the realm of weather not climate – clarify the difference between the two and really what the important processes are.

Amended. We now refer to days and weeks as synoptic conditions.

~L322: Temperature might not be driving melt, but something in the model clearly is driving melt. Look at other parameters to identify this. For example, is it wind that could be causing it? That would be logical, and really helpful to identify the driver of such an important process.

We now link melt events to atmospheric circulation anomalies (see section 5.2).

L332: It may be the first time this has been published explicitly – but it is also not surprising. There are a few papers that observe supraglacial lakes on East Antarctic outlets. So why is it important that this has been observed for the first time?

We now clarify the importance of surface melt on sea-ice. In the arctic, this has been linked to sea-ice break-up. Therefore, the fact that we observe ponding on the multi-year sea-ice prior to its fracturing and ultimate break-up suggests that surface melt could have been important.

~L347: Is it possible that the higher melt year saturated/refroze in the snowpack, which then allowed a lower melt year to be able to form melt ponds? I know that is the case on ice shelves, but I'm not sure if that is true in a sea ice context?

This is an interesting point. Studies in the Arctic suggest the trapped latent heat within the sea-ice as melt ponds refreeze may inhibit basal growth of the sea-ice (e.g. Flocco et al., 2015; JGR). We now include this point in the manuscript.

~L362: The sentences around here go in a couple circles about the processes and drivers that you think are most important for the reader to understand. I think it might help to clarify that, in this system, bathymetry and geometry seem to drive the location of calving events which sea ice drives the timing.

Section removed

L376: This is restating earlier conclusions. Maybe only need to say in one place?

Section removed

L380-389: This is really interesting and important glaciologically! The paper would be stronger if this were fleshed out and done so with more rigor. It can very much be a discussion of what is reasonable (not an in-depth analysis), but more should be included. For example, what might changed in accumulation do? Is it possible basal changes played a role? What else could be driving increased velocity?

Section removed

~L399: Yes, sea ice is related to climate – but Antarctic sea ice is very much dependent on more than temperature (which can be seen in regional expansion of Antarctic sea ice). More nuance needs to be brought to this sentence.

The conclusion has been modified to reflect the greater discussion on atmospheric circulation anomalies. However, we note that multi-year landfast ice may to respond to a different set of climatic drivers to sea-ice extent, which has been increasing.

L409: You specifically mention "warming" – but it would seem to be more appropriate to discussions in atmospheric or oceanic circulation?

We now discuss in detail the anomalous atmospheric circulation patterns.

L411: Okay – but where else might these processes be important? Expand this conclusion to be broader to have larger implications.

The conclusion has been expanded.

Table 2: No Landsat 8 OLI imagery used? This might be interesting for the recent breakup and data are available from 2013.

The 2016 event started in late March. Therefore, Landsat 8 only made a few passes before the polar night and most of the time it was cloudy.

Figure 1: **Include a small inset of the entire continent. **"Moscow University" should be "Moscow University Ice Shelf" **Scale bar in upper figure

We have removed figure 1a and added a small inset to Figure 1 (previously 1b).

Figure 2: **x-axis labels are a little too small **Caption should note the different vertical scales

Amended

Figure 3: **Blue is a bit hard to see **Show outline of this area in Figure 1? Don't worry about it if too crowded. **Brighten figure so easier to view

Blue has been changed to yellow and the figure has been brightened.

Figure 4: **Consider tracing front in a 2nd color in each image to clarify the changes that you want to highlight between images? It is hard to see (as you admit) with the melt, etc. **You reference the total area calved. Maybe include a hatched area in the last image between the two terminus lines?

It is very difficult to digitize the front in each image because of the abundance of icebergs close to the terminus during the calving event.

Figure 6: Increase brightness and contrast to make more easily viewable.

Amended

Figure 7: Anomalies are interesting but is an absolute scale better to demonstrate what you want show?

An absolute scale is very difficult to follow due to the seasonal variations. Therefore, we have stuck with anomalies.

Figure 8: Include 2nd outline in lower image?

Figure removed

Figure 9: **Maybe darken a little so it prints better? **Include 2nd outline in lower image?

Figure removed

Figure 10: Increase contrast so more viewable. The edge of the 9 km advance isn't very visible when printed.

Figure removed

Figure 11: Increase contrast in lower image.

Figure removed

Figure 12: Same comment as in text – include inferred velocities for these time periods, too?

This figure is now used to simply indicate the estimated terminus position of Holmes (West) Glacier, see section 4.6.

Figure 13: Add line for 7 Feb? Hatched area to indicate calved area?

Amended. Line added and figure updated with more recent imagery.

Figure 14: **Include annotation in each image and particularly at circle to help the reader **Increase contrast to make more viewable.

Further description has been added in the figure caption. Contrast has been increased.

Figure 16: Maybe just report January '14 melt total relative to 2007? I don't think that the timeseries is particularly helpful here.

Figure removed

Figure 17 & 18: Combine these into one figure? Figure 18: **Be consistent with date format **Double check permissions and copyright for using a Google Earth image in this publication.

Figure removed. See new figure 13
* * *
**Reviewer 2**

This is an interesting but rather rambling study on the connection between the presence of fast ice and the timing of glacier calving, and its potential impact on glacier flow. The study uses MODIS image data and the record of sea ice concentration from SSM/I (mostly) to demonstrate that during the brief periods of sea-ice-free conditions in Porpoise Bay, significant calving occurs, implicating the fast ice as a stabilizing component. The study then continues to search backward in time for evidence of this relationship, to the earliest satellite data, and then forward in time to an event in early 2016.

We thank the reviewer for appreciating that our study is interesting.

It could be published as it is. . . it is not incoherent. But it does not offer a clear contribution beyond the initial worthwhile documentation that fast-ice break-up leads to rapid iceberg release; or, to say it conversely, the presence of fast ice inhibits calving and drift. The discussion of velocity change seems rather vague, since any speed-up is not able to be directly tied to fast ice break-out and increased calving. The link to surface melting is so tenuous as to be useless. Moreover, the writing style is not brisk and efficient. There is a lot of excess text.

We are pleased that that the reviewer suggests that despite some reservations our study could be published 'as it is'. We disagree that the link to surface melt is 'useless'. We agree that RACMO won't capture katabatic heating events, but we note that there are no weather stations within 100s km of Porpoise Bay, so it is arguably the best dataset available. We now present new imagery from early December 2005 and mid-January 2006, which shows the development of fractures in the landfast ice and link this larger scale atmospheric circulation anomalies. These same fractures eventually rupture initiating the sea-ice break-up in 2007. Therefore, the synoptic conditions throughout December 2005 are likely to have been important in driving the break-up of sea-ice and subsequent glacier calving in 2007, which we now discuss in more detail. During December 2005 we observe surface melt features on the landfast ice. Given that the RACMO model implies that the mean December 2005 melt was exceptional for the region, we suggest that surface melt (along with other associated processes) may have been important process in weakening the landfast ice prior to break-up. To our knowledge no other studies have considered surface melt in landfast ice break-up in Antarctica. We simply highlight that this could be an important process and is worth considering.

The central discovery of the study appears to be a major glacier calving event in late summer of 2007, during a period of extended sea ice retreat in Porpoise Bay. Images before, during and after the break-out of the sea ice show the disaggregation of the floating ice tongues in the region. Having found this, the authors extent the search, first backward in time through the sea ice record, and then back still further using the record of early Landsat and declassified Argon / Corona data.

There is another data set available, already processed, of MODIS data. See http://nsidc.org/data/iceshelves_images/index_modis.html This is a processed geolocated record of ~monthly to weekly images going back to 2000. This shows other periods of low sea ice and partial fast ice break-up. Unfortunately, there is a one-year gap around 2001-2002 that could help narrow down the 2002 calving. If you request some additional images, we can add them for this time range.

This manuscript is very long, and could be much better focused. It is not necessary to show every data set that can say something about the 2007 event – the purpose is to document the event and the link with fast ice break-out, and then examine the extent to which glacier flow might change as a result. In particular, it is lengthy to read, first, the discovery of the 2007 event, then the inference of the 2002 and and 1986 events, and then the possibility that there was no calving between 1963 and 1973 (and that case is not well-made) and then that there was a recent event.

The paper should present what you've learned overall, not the step-by-step process by which you learned it.

I don't want to spend more time with the study. Nor do I want to berate the writers. So let me simply outline the paper they should write, if they wish to. Figure order is re-arranged. Any figure not listed is not needed.

Introduce the region (Figure 1b) Indicate the location of later figures here. Present the evidence for major break-ups in March 2016 and February 2007: (fig 13);(merge Figure 3-dates?- into Fig 4);(then, Fig2 extended through 2016). Fig6 might be retained, but it does not really help - regional sea ice is not low, yet there is a calving in austral autumn 2006.

Present the sea ice record from SSM/I+SMMR+EMSR for the areas in Fig 1b (using Figure 7) The major bay-wide calving and retreat is clearly timed by the loss of sea ice.

Earlier events large events at the Holmes Glacier front can also be linked, somewhat tentatively to extreme sea ice lows. MODIS data at NSIDC could help greatly here. An examination of this image series could document that the small iceberg pattern does / does not remain fixed (spreading or twisting of the arrangement of the bergs, but no 'individual' motion) during periods of continuous fast ice. Create a new figure from the MODIS data, summarizing what it shows about the small berg pattern evolution over time (this has a bit of Fig 14, but might be more extensive). Don't have to show every event in the 2000-2016 record – just the facts you extract from it.

With this record, i.e. ASAR, sea ice concentration, and MODIS, you have observations that can help you interpret the old record. Create a figure of the 1997, Nov 1973, Jan 1973, and Oct 1963 images, together. Given the field of small bergs trapped in fast ice in this series, make inferences about past calvings of Holmes, and of Frost (Figure 12). I would downplay the calvings of the other small glaciers, it is too confusing to follow it all.

From here, discuss synoptic patterns using reanalysis data that cause (or are associC3 TCD Interactive comment Printer-friendly version Discussion paper ated with) sea ice retreat from this region of Antarctica. I don't think surface melting helps much, and in any case RACMO is not likely to

capture katabatic heating events very well, which could be key in the immediate vicinity of the grounding line. I don't see how Fig 15, 16, 17 or 18 really helps.

This will be a new paper, that you would re-submit for review. It would be half as long. It would conclude things about calving and presence of fast ice, mostly focused on the the 2016, 2007, and 2002 events, but with a few statements about earlier events back to 1963. It would conclude things about berg motion within temporally continuous fast ice, and synoptic climate patterns that favor fast ice break-out and calving in this area.

The manuscript has been shortened and re-structured broadly along the lines the reviewer suggests, but also keeping in mind that Reviewer 1 specifically noted the helpfulness of the Figures and requested additional discussion. To summarise: we have removed the lengthy discussion on reconstructing the calving cycles and now focus on the drivers of sea-ice break-up in 2007 and 2016. We link the 2007 event to atmospheric circulation anomalies weakening the sea-ice in December 2005 prior to its break-up. In contrast, we find no link between atmospheric anomalies and the 2016 event sea-ice break-up. Instead, we link this event the terminus positon of Holmes (West) Glacier, pushing the multi-year sea-ice further into the open ocean. As noted in the revised conclusions, despite these different mechanisms, our manuscript clearly demonstrates the importance of landfast sea-ice on major calving events in East Antarctica.

We have reduced the number of Figures, but given that the satellite imagery are the primary source of data for this study, we feel it is very important to show these data on Figures. We also note that Reviewer 1 specifically praises this aspect of the manuscript "*This paper includes a lot of figures which help demonstrate and illustrate the arguments in the paper. This is really helpful!*"

---

## Referee Report (RR1)

[referee-annotated manuscript omitted]

---

## Author Response (AR2)

We thank the reviewer for once again taking the time to provide feedback on our manuscript. We include a reply to the two minor points raised below and have edited the text and figures accordingly to the small comments in the attached PDF:

*1) The sections which discuss the role that atmospheric anomalies play in the break-up are a little bit difficult for me to follow. I hope the authors consider re-writing the sections so that they are a little easier for the reader to follow. Specifically, use simpler sentences which follow each other and smaller paragraphs each with a defined goal.*

We have edited and re-written this Section (4.4) to make it clearer to follow.

*1.1) On a related topic, some of this material is repeated in section 5.2. It is hard to tell if this is an oversight or intentional. Please revisit to be sure.*

We have tried to reduce any obvious repetition (e.g. through simplifying and shortening the results in Section 4.4), but some discussion of these results inevitably leads to some reiteration of the key points in section 5.2. This is intentional.

*2) While the atmospheric conditions described in the paper are helpful for explaining the break-up events, I believe each requires a little more attention. In particular, prior to the first calving event, atmospheric circulation clearly weakens the sea-ice, when is there a long lag before the break-up event? I think this needs to be better explained. And for the second event, is there any information which corroborates your hypothesis that this is the outer limit of sea-ice stability, except for the repeated location of calving? There are many geographical factors which could explain that.*

The warm anomaly in December 2005 was only short lived and by January 2006 conditions returned back close to average and remained so until the austral winter, where sea-ice break up is less likely. This may explain the lag between the onset of sea-ice fracturing and its eventual break-up in the following summer. We have added this to section 5.2.

Regarding the calving event in 2016: we agree that there are many geographical factors which could explain why Holmes (West) Glacier calves in roughly the same position in each cycle. However, we note that the bathymetry in Porpoise Bay has yet to be surveyed, which makes further discussion on this point purely speculative.

[revised manuscript text omitted]